# High-Temperature Oxidation of High-Entropic Alloys: A Review

**DOI:** 10.3390/ma14102595

**Published:** 2021-05-16

**Authors:** Sergey Veselkov, Olga Samoilova, Nataliya Shaburova, Evgeny Trofimov

**Affiliations:** Department of Materials Science, Physical and Chemical Properties of Materials, South Ural State University, 76 Lenin Av, 454080 Chelyabinsk, Russia; mesved123@yandex.ru (S.V.); samoylova_o@mail.ru (O.S.); tea7510@gmail.com (E.T.)

**Keywords:** high-entropic alloys, high-temperature oxidation, HEAs of Al-Co-Cr-Fe-Ni system, HEAs of Mn-Co-Cr-Fe-Ni system, Refractory HEAs

## Abstract

Over the past few years, interest in high-entropic alloys (HEAs) has been growing. A large body of research has been undertaken to study aspects such as the microstructure features of HEAs of various compositions, the effect of the content of certain elements on the mechanical properties of HEAs, and, of course, special properties such as heat resistance, corrosion resistance, resistance to irradiation with high-energy particles, magnetic properties, etc. However, few works have presented results accumulated over several years, which can complicate the choice of directions for further research. This review article presents the results of studies of the mechanisms of high-temperature oxidation of HEAs of systems: Al-Co-Cr-Fe-Ni, Mn-Co-Cr-Fe-Ni, refractory HEAs. An analysis made it possible to systematize the features of high-temperature oxidation of HEAs and propose new directions for the development of heat-resistant HEAs. The presented information may be useful for assessing the possibility of the practical application of HEAs in the aerospace industry, in nuclear and chemical engineering, and in new areas of energy.

## 1. Introduction

A promising area of development in the field of material science is research on the creation and properties of high-entropy alloys (HEAs).

The first works on HEAs were presented by scientists from the University of Taiwan [1,2,3]. Based on the assumption that a high value of the configurational entropy of mixing can help suppress the formation of intermetallic phases and stabilize a multicomponent single-phase solid solution of metals, it was found that a single-phase structure consisting of a simple solid solution is formed in alloys of equimolar composition formed using five or more metals. Since it was assumed that the phenomenon is associated with a high entropy of mixing, it was proposed to name alloys consisting of five or more elements in approximately equal equimolar concentrations “high-entropy alloys”.

Further investigation of their properties showed that such multicomponent alloys are very hard, wear resistant, heat resistant, corrosion resistant and have good low-temperature plasticity and superplasticity [4,5,6].

Five main groups of HEAs can be distinguished [7,8] based on:(1)metals of the Fe subgroup and similar transition metals—Fe, Co, Ni, Cr, Mn, Cu;(2)transition refractory metals—Ti, Zr, Hf, Nb, Ta, Mo, W;(3)platinoids and Au;(4)light elements—Al, Li, Sc, Mg, Ti, Si;(5)lanthanide alloys.

From the point of view of the use of alloys, the study of their high-temperature oxidation (in the context of studying their high-temperature corrosion resistance) is of particular interest. The study of HEA oxidation is useful for analyzing the formation of HEAs and analyzing their high-temperature gas corrosion behavior. The potential for using HEAs in high-temperature environments is significant; however, there is no understanding of their ability to oxidize or development of effective models to predict their oxidative behavior. A number of studies have summarized the most diverse aspects of the formation of HEAs [7,8,9,10,11,12], but practically none has been devoted to the oxidation of HEAs at elevated temperatures (despite the fact that a fairly large amount of information has been obtained in this area). One exception is a paper by Chen et al. [13], where only some aspects of high-temperature oxidation of refractory HEAs are considered (moreover, a sufficient amount of new data has been published over the past three years). The Ellingham diagram [14,15], which shows the change in the Gibbs energy of the formation of metal oxides depending on the temperature and composition of the gas phase, can give some idea of the possible oxidative behavior of HEAs at high temperatures.

HEAs based on light elements and lanthanides, taking into account the properties of the elements that form them, are oxidized easily and completely. On the other hand, the possibility of high-temperature oxidation of platinum-group metals at pressures close to atmospheric is hardly worth studying. If we analyze the data of the Ellingham diagram [14,15], as well as the data of the Pilling-Bedworth ratio [16] (which is a criterion for the continuity of the formed oxide film), we can conclude that interesting data of oxidation can be observed in HEAs based on metals of the Fe subgroup and HEAs based on refractory transition metals. Therefore, in this review, we analyze works devoted to the high-temperature oxidation of these groups of HEAs. It should be noted that among the systems based on the Fe subgroup, the most studied are the Al-Co-Cr-Fe-Ni and Mn-Co-Cr-Fe-Ni systems. It is possible that this is due to the fact that the Al and Cr in their composition, which, according to the Ellingham diagram, are oxidized more readily than other components of these alloys (i.e., the formation of oxides of these elements is thermodynamically more likely to occur) and have a Pilling-Bedworth ratio more than one, impart protective properties upon films made of their oxides.

The relevance of preparing a review on this topic is provided by their potential practical application in the aerospace industry, in nuclear and chemical engineering, in new areas of energy, and for applications in which the resistance of alloys to oxidation is an extremely important characteristic. In addition to creating new heat-resistant materials (for example, for gas turbine blades operating at elevated temperatures), research in this area will help substantiate the composition of corrosion-resistant coatings, including by additive technologies. Such coatings can be used in aircraft and rocketry, the nuclear industry and chemical engineering. Of particular interest are the prospects for creating oxidation resistant HEAs for use in the nuclear industry for the latest thermonuclear reactors, and generation IV nuclear reactors. It is necessary to develop new materials with increased thermal and structural stability, a high level of mechanical properties, and a higher resistance to corrosion and radiation in comparison with the structural materials used in generation II and III reactors [17,18,19,20].

This review consists of six chapters. The first introduces the problem. The second and third chapters are devoted to a review of the oxidative behavior of HEAs based on the Fe subgroup; the most common and studied systems of HEAs are Al-Co-Cr-Fe-Ni and Mn-Co-Cr-Fe-Ni, respectively. The systems were divided into separate chapters, not only for the convenience of the reader, but also because of a fairly large amount of literature data on each of the systems. The fourth chapter is devoted to a review of the behavior during high-temperature oxidation of HEAs based on refractory metals. The fifth chapter summarizes the data on the oxidation products of all considered alloys, and also considers the possibility of predicting the phase composition of the products of high-temperature gas corrosion using thermodynamic modeling techniques. The need to include data in the fifth chapter is due to the accepted convenience of reading the review article and the use of data from different sources. The sixth chapter is a logically built chain of conclusions, taken on the basis of all the considered literature data on high-temperature oxidation of HEAs.

## 2. Al-Co-Cr-Fe-Ni System

In conventional alloys, oxidation resistance can be greatly improved by the addition of a suitable (usually small) amount of Al, Cr or Si, as these elements can form a dense and stable oxide layer on the surface at high temperatures [21]. HEAs based on metals of the Fe subgroup have fewer compositional restrictions than conventional structural alloys (e.g., stainless steels, Ni-based alloys), and may contain higher concentrations of elements such as Al, Cr and Si, which are necessary to form external protective oxide films [22]. Accordingly, such HEAs have a potentially higher resistance to high-temperature gas corrosion. It is also known that HEAs of this group are characterized by a low diffusion rate of elements [23,24,25,26], which complicates the formation of oxide particles in the bulk of these alloys.

Among HEAs based on metals of the Fe subgroup, the Al_x_-Co-Cr-Fe-Ni system is one of the most thoroughly studied [25,27,28,29,30,31,32,33]. Most of the articles presented in the literature describe the high-temperature oxidation of HEAs of this group in air, with the exception of the work of Liu et al. [27], which describes the corrosion resistance of alloys at high pressures in an atmosphere containing water vapor. Since there is practically no data in the literature on the behavior of HEAs under such conditions, we are obliged to dwell on the description of work [27] in more detail.

Liu et al. [27] studied the oxidative behavior of high-entropy Al_x_CoCrFeNi alloys (x = 0.15; 0.4) in supercritical water. In ultra supercritical pressure (USP) boilers, due to the increased temperature and pressure, an aggressive environment is created that adversely affects boiler materials. For materials working in such an environment, a high level of heat resistance is required. Liu et al. suggested that HEAs are a material that could be used in USP boilers. The alloys Al_0.15_CoCrFeNi (Al_0.15_) and Al_0.4_CoCrFeNi (Al_0.4_) were obtained by vacuum induction melting and casting, after which the oxidation of the alloys was carried out at temperatures of 550 °C and 600 °C and pressures of 23 MPa and 25.5 MPa for 70 h, respectively.

For comparison, oxidation of HR3C alloy [27] was also carried out at 600 °C and 23 MPa for 70 h (heat-resistant steel HR3C (25Cr20NiNbN) is a typical alloy used for the manufacture of pipes for the superheater and preheater in USP boilers). A comparison of the surface morphology of the Al_0.15_, Al_0.4_, and HR3C alloys after oxidation is shown in Figure 1. The oxide films formed on the Al_0.15_ and HR3C alloys have a two-layer structure, where the outer layer consists of Fe_3_O_4_, and the inner layer consists of FeCr_2_O_4_, while the oxide film on the Al_0.4_ alloy mainly consists of a single layer of spinel FeCr_2_O_4_. Compared to HR3C steel, thinner oxide layers with a small oxide particles were obtained on the Al_0.15_ and Al_0.4_ alloys (Figure 1 and Figure 2), which indicates the noticeably higher heat resistance of these alloys compared to HR3C steel.

Among other data presented in the literature on high-temperature gas corrosion in air for the HEAs based on the Al_x_-Co-Cr-Fe-Ni system, the most complete study, in our opinion, was carried out by Butler et al. [29,30,31].

Butler and Weaver [29] studied the microstructure and oxidation mechanisms in a series of HEAs based on AlCoCrFeNi. Alloys of composition Al_x_(CoCrFeNi)_100-x_ (where x = 8; 10; 12; 15; 20; and 30 at. %) were obtained by melting pure components (purity > 99.9%) in an arc furnace on a water-cooled Cu hearth in an Ar atmosphere. The oxidation of HEAs samples was carried out at 1050 °C in air in a tube furnace for 100 h. After the experiment, the structures of the obtained alloys were studied. The authors found that the isothermal oxidation of HEAs in air at 1050 °C promoted the formation of films of Cr_2_O_3_, Al_2_O_3_, NiCr_2_O_4_, as well as a separate AlN phase. For HEAs with a low Al content (Al_8_ and Al_10_) an outer layer consisting of Cr_2_O_3_ was formed simultaneously with an inner non-uniform layer of Al_2_O_3_ (Figure 3a). For HEAs containing 12% and 15% Al (Al_12_ and Al_15_) simultaneously with Cr_2_O_3_, an outer layer of NiCr_2_O_4_ spinel was formed, followed by an inner discontinuous layer of Al_2_O_3_, and AlN precipitated out as a separate phase (Figure 3b). For HEAs with a high Al content (Al_20_ and Al_30_), an outer layer consisting of Cr_2_O_3_ was formed together with an inner non-uniform layer of Al_2_O_3_ and a small amount of the precipitated AlN phase (Figure 3c). In general, an increased Al content has been shown to increase the oxidation resistance by promoting the formation of a continuous Al_2_O_3_ film. The work demonstrated that the components of compositionally complex alloys can be oxidized selectively and lead to the formation of protective oxide layers.

Next, Butler and Weaver [30] studied the effect of annealing on the structure and phase composition of the alloys listed in [29], and on the oxidative behavior of these alloys after annealing in comparison with the oxidative behavior of cast alloys from [29]. It was found that the order of magnitude of the change in mass between the corresponding HEAs in the cast and annealed state was almost the same. However, the concentration of Al in the samples of cast HEAs has a much greater effect on oxidation than for the annealed samples of the same compositions. In both cases, an increase in the Al content leads to a decrease in the numerical readings of the weight change curves. In all cases, annealing stabilized the microstructure and promoted the formation of an external continuous film α-Al_2_O_3_. The annealed HEAs also exhibited longer adherences to protective, parabolic oxide growth.

Butler et al. [31] studied the oxidative behavior at 1050 °C of the HEAs Al_2_Co_4.5_Cr_4.5_Fe_4.5_Ni_4.5_ and Al_4_Co_5_Cr_5_Ni_5_Si, and compared it with the results of oxidation at 1050 °C of the heat resistance alloy Al_3_Co_7_Cr_2_Ni_7_Si, which is a high-temperature shape-memory alloy [34,35]. They showed that the oxidation of HEAs with an Al content of 10 at. % leads to the formation of an outer layer of Cr_2_O_3_ and an inner inhomogeneous discontinuous layer of Al_2_O_3_ together with AlN inclusions. The result was that the weight gain of this alloy increased markedly during testing. Alloys with 15 and 20 at. % Al formed continuous layers of Al_2_O_3_ with a small fraction of Cr_2_O_3_. When testing alloys with such Al concentrations (15 and 20 at. %), a smaller weight gain was observed, which confirmed the protective properties and the continuity of the formed layer.

It is possible to increase the resistance to oxidation of the HEAs of the Al-Co-Cr-Fe-Ni system by introducing additional alloying elements. The most interesting results on doping the base alloy with copper were obtained in [32,33].

Dabrowa et al. [32] studied the effect of Cu on the oxidation ability of the HEAs AlCoCrCu_x_FeNi. Three different HEAs were obtained by melting in an arc furnace: AlCoCrFeNi, AlCoCrCu_0.5_FeNi, and AlCoCrCuFeNi. Their crystal structures were bcc, bcc and bcc + fcc, respectively. All the investigated alloys were oxidized in air in a quartz reactor at 1000 °C for 100 and 500 h. This treatment led the AlCoCrFeNi alloy with the bcc lattice changing to two-phase structure (bcc + fcc). This was due to depletion of the surface layers of the metal directly under the oxide layers by Al due to its oxidation. Conditions were created for the formation of an alloy layer with an fcc structure. The treatment of the AlCoCrCu_0.5_FeNi alloy with the bcc lattice was accompanied by similar processes. Small precipitates of the third Cu-rich phase were also observed. After 500 h of treatment, part of the fcc phase transformed into the σ-phase (Fe, Co)Cr, which was accompanied by the growth of grains with a high content (up to 80 at. %) of Cu. The behavior of the high-entropy AlCoCrCuFeNi alloy during oxidation was very similar to AlCoCrCu_0.5_FeNi, with the most notable difference being the higher content of the Cu-rich phase. During annealing, it was noticeable that the Cu content in this phase increased from 55 at. % (after 100 h of treatment) up to 80 at. % (after 500 h of treatment). Scanning electron microscope (SEM) images of alloy structures and the points chemical composition of EDS (electron microprobe analysis performed on an energy dispersive spectrometer) after the experiment are shown in Figure 4.

For the AlCoCrFeNi alloy, after 100 h of exposure at 1000 °C in air, the main product of high-temperature oxidation was Al_2_O_3_ (see Figure 4a), but after 500 h at this temperature, the oxide layer became multiphase, consisting of Al_2_O_3_ and phases based on the spinels FeCr_2_O_4_ and FeCo_2_O_4_ (see Figure 4d). For the AlCoCrCu_0.5_FeNi alloy, after 100 h of exposure at 1000 °C (see Figure 4b), the composition of oxidation products is characterized by the presence of both Al_2_O_3_ (in the form of a separate phase) and oxides (probably spinels) containing all the main components of the alloy, and it should be noted that the Cu in these oxides contains the smallest amount of all metals (no more than 2 at. %). After 500 h of exposure at 1000 °C in air for this alloy (see Figure 4e), the scale composition changes towards a higher Cu content (up to 10 at. %). For an alloy with a higher Cu concentration AlCoCrCuFeNi, after 100 h of exposure (see Figure 4c), the phase composition of the oxide layer is similar to that of the AlCoCrCu_0.5_FeNi alloy. After 500 h of exposure (see Figure 4f), only Al_2_O_3_ is included in the scale.

Dabrowa et al. [32] determined that with an increase in the Cu content, the adhesion of the oxide film with respect to the alloy surface deteriorated. That is, for the AlCoCrFeNi alloy (without Cu), the oxide film on the surface adhered relatively well; for the AlCoCrCuFeNi alloy (with Cu), almost all the oxide films peeled off the surface.

Daoud et al. [33] studied the oxidative behavior of Al_0.5_CoCrCu_0.5_FeNi_2_, Al_1.5_CoCr_1.5_Cu_0.5_FeNi and AlCoCrCuFeNi (Al_0.5_, Al_1.5_ and Al_1_) at high temperatures in air. After being treated at 800 °C for 200 h, all alloys showed little weight change. Three layers of oxides appeared on the surface of the Al_1.5_ alloy sample, including Al_2_O_3_ (inner oxide layer), then Cr and Ni oxides (middle layer), and then Fe oxide (outer layer). Al_0.5_ alloy showed a higher weight gain at 1000 °C, while oxide layers based on Cr oxide and aluminum oxide were formed on the surface. Detachment of the oxide layer from the surface was observed on samples of alloys Al_1_ and Al_1.5_ at 1000 °C.

Alloying with highly active elements (for example, Y or Hf) in conventional alloys (for example, NiCoCrAl, NiAl, or FeCrAl alloys) can significantly improve oxidation resistance [36,37,38,39,40,41]. Considering this fact, Lu et al. [28] alloyed AlCoCrFeNi with Y and Hf (0.02 at. % each). The isothermal oxidation experiment was carried out at 1100 °C in a chamber furnace in air for times ranging from 1 to 1000 h. This alloy had excellent oxidation resistance while maintaining high mechanical strength and heat resistance at high temperatures. The oxide layer formed on the AlCoCrFeNi with Y and Hf (0.02 at. % each) mainly consisted of α-Al_2_O_3_ grains, at the boundaries of which small amounts of oxides with alloying elements Y and Hf (for example, Y_3_Al_5_O_12_ and HfO_2_) were found—see Figure 5. This layer was uniform in thickness, and after 1000 h at 1100 °C, had a thickness of only 4.6 ± 0.3 μm. The oxide/alloy interface was smooth and clean with no surface defects.

The data of different authors on the determination of the weight gain during the oxidation of HEAs based on the Al-Co-Cr-Fe-Ni system are summarized in Table 1. The summary information given in Table 1 is supplemented with the activation energy calculated by us (data used for holding for 20 h). 

After analyzing the data from Table 1, it can be noted that the best results for the base alloy were obtained for the composition Al_30_(CoCrFeNi)_70_ [29]; copper additives increase corrosion resistance (AlCoCrCuFeNi composition [32]); and the alloy with the addition of silicon and without iron in its composition showed the greatest resistance (Al_3_Co_2_Cr_7_Ni_7_Si [31]). At the same time, all three alloys are characterized by a protective oxide film based on Al_2_O_3_ with the possibility of forming a layer of Cr_2_O_3_. It should also be taking in account that in the work of Lu et al. [28] there is an extremely low corrosion rate of the alloy AlCoCrFeNiYHf (kp = 1.9 × 10^–13^ g^2^ cm^–4^ s^–1^), which probably explains the lack of data on weight gain. Thus, the additions of yttrium and hafnium have a positive effect on the corrosion resistance of HEAs based on the Al-Co-Cr-Fe-Ni system.

It can be noted that literature data are mainly of an empirical nature, and only represent the results of experiments in the form of a technical report. As such, there is no analysis of the mechanism of the effect of adding various elements to the base alloy Al-Co-Cr-Fe-Ni on the process of high-temperature gas corrosion. In this connection, the question arises about the gaps in the presented approaches in the study; in particular, it is necessary to use the tools of thermodynamic modeling to describe the dependence of the phase composition of the formed corrosion products from the alloy composition and oxygen pressure in the system. There is also insufficient data on the kinetic laws of high-temperature oxidation of HEAs based on the Al-Co-Cr-Fe-Ni system.

Thus, there is no clearly defined line of research in the literature. The influence of hafnium additives seems promising, which leads to the idea of a possible positive effect of tantalum or tungsten additives. It also makes sense to consider the complex effect of doping with copper in conjunction with gold, silver and platinum (according to the data from Ellingham diagram).

## 3. Mn-Co-Cr-Fe-Ni System

Another HEA system, which has also been repeatedly investigated in terms of the mechanisms and results of its oxidation, is the Mn-Co-Cr-Fe-Ni system, because the diffusion coefficients in MnCoCrFeNi alloys are lower than in traditional alloys with an fcc lattice [42]. The resistance to high-temperature gas corrosion of HEAs this group of alloys was studied most thoroughly in [42,43]. Let’s dwell on the results of these works in more detail. Holcomb et al. [42] conducted an experiment on the oxidation in air of eight HEAs MnCoCrFeNi (HEA-1 to HEA-8 alloys, see Table 2) for 1100 h at 650 °C and 750 °C.

The authors studied the kinetics of oxidation by measuring the weight gain of the samples. Five HEAs were alloyed with small amounts of Nb and Cr to improve the strength of the alloy. Six HEAs were alloyed with Y in an attempt to reduce the S levels in the alloys and to increase oxidation resistance. In Cr-containing systems of HEAs, according to the authors, Cr should have provided a higher resistance to oxidation in comparison with systems without it. The oxidation properties of these HEAs were compared with Ni Superalloy 230 and 304H Austenitic Stainless Steel.

The results of the change in mass are shown in Figure 6a,b for 650 °C and 750 °C, respectively. The best resistance to oxidation at both temperatures was found in the alloys 230, 304H and HEA-1, which had a low overall weight gain. Neither alloy 230 nor HEA-1 had traces of delamination. Alloy 304H showed slight weight loss after 800 h at 750 °C.

A thin oxide film on CoCrFeNi (HEA-1) mainly consisted of Cr oxide (presumably Cr_2_O_3_) with a small amount of Mn oxide on the outer part of the film. All MnCoCrFeNi alloys (from HEA-2 to HEA-7) underwent faster oxidation than HEA-1. In this case, especially at 750 °C, peeling of the oxide film was observed. The thickness of the oxide layer was greater for these alloys in comparison with HEA-1; the outer layers of the scale were enriched in Mn, and the inner layers (near the alloy/scale interface) were enriched with Cr oxides. HEA-6 and HEA-7 alloys, containing more Fe and less Cr than other alloys of this series, had a different scale structure (Figure 7). The oxide film for these two alloys was also multilayer, consisting of different oxides: the outer and inner (at the interface with the metal) oxide layers were rich in Mn, and the inner layers of oxide film were enriched in Fe. The Cr content in the layers gradually decreased moving from the matrix to the outer oxide layer. Comparing the alloys HEA-2 to HEA-7, the HEA-2 alloy demonstrated a higher oxidation rate and a greater degree of exfoliation of the oxide layer. This difference in oxidation characteristics is explained by the authors by the absence of Y and/or an increased amount of sulfur in HEA-2. It has been shown that the addition of Y, mainly to reduce the amount of S in the alloy, also improves the oxidation resistance of these HEAs, most likely by improving the adhesion of the oxide layer to the alloy.

MnCoFeNi (HEA-8) demonstrates much higher oxidation and scale delamination rates than Cr-containing alloys (Figure 8). Thermodynamic calculations showed that in alloys with Cr, in which spinel MnCr_2_O_4_ was formed at the alloy surface/oxide layer interface, internal oxidation of Mn is prevented, while in HEA-8 the formation of MnFe_2_O_4_ at the alloy surface/oxide layer interface does not prevent further internal oxidation of Mn.

The main conclusion of [42] is that if HEAs of the Mn-Co-Cr-Fe-Ni system are used in high-temperature oxidizing environments, one should focus on the use of alloys characterized by a sufficiently low Mn content and a high Cr content.

Laplanche et al. [43] focused on the study of the properties of the Mn_20_Co_20_Cr_20_Fe_20_Ni_20_ alloy (composition in at. %), (MnCoCrFeNi). They believed (referring to [44]) that when this HEA undergoes oxidation, Mn and Cr oxides are formed, rather than Fe, Co or Ni oxides, since the Gibbs free energies of formation of the first oxides are more negative than the latter. In addition, the authors proceeded from the fact that the presence of Mn, which is used as a stabilizer of the austenite phase in austenitic steels, has a detrimental effect on the oxidation resistance. This is due to the spinel enrichment of Mn in the oxide layer, since the diffusion rate of metal cations through the Mn spinel is higher than through the protective layer of Cr oxide. Wild [45] reported that the transition of Mn from metal to oxide film leads to a decrease in the proportion of Mn near the oxide/metal interface. Since the diffusion of Mn in MnCoCrFeNi is slow [23], the authors were interested in determining whether this stage (and not diffusion through the oxide layer) controls the overall rate of oxidation. To this end, the authors subjected MnCoCrFeNi to isothermal oxidation in air at temperatures from 500 °C to 900 °C for different periods of time up to 100 h.

Thermogravimetric analysis of high-entropy MnCoCrFeNi alloy in [43] showed an initially linear oxidation rate, which became parabolic over longer periods of time. The Arrhenius plot for the parabolic rate constant gave an activation energy of 130 kJ/mol, which is comparable to the diffusion of Mn in Mn oxides (~122 kJ/mol) [46], but much lower than at the rate of diffusion of Mn in HEA (~288 kJ/mol) [23]. The diffusion of Mn through the oxide is a rate-limiting process [23]. At 600 °C, the oxide film consisted mainly of α-Mn_2_O_3_ with a thin Cr_2_O_3_ layer near the interface between oxide and matrix. α-Mn_2_O_3_ was retained up to 800 °C, but converted to Mn_3_O_4_ in the temperature range between 800 °C and 900 °C. As a result of oxidation, the metal base of the MnCoCrFeNi alloy was partially depleted in Cr and Mn, while pores were formed near the oxide film (Figure 9).

The data of different authors on the determination of the weight gain during the oxidation of HEAs based on the Mn-Co-Cr-Fe-Ni system are summarized in Table 3.

Analyzing the data from Table 3, we note that the alloy which showed the best corrosion resistance was Mn_0.5_Co_26_Cr_22_Fe_25_Ni_26_ [42], whose oxide film consisted only of Cr_2_O_3_. This is consistent with the calculation of the activation energy, according to which the rate of the oxidation reaction for this alloy will be the lowest of all. Yttrium additions did not affect on the corrosion resistance of the presented alloys.

Mn-Co-Cr-Fe-Ni alloys seem less attractive in terms of their corrosion resistance than the alloys in the previous chapter. Comparing Table 1 and Table 3, it can be noted that even the test temperature for Mn-Co-Cr-Fe-Ni alloys is lower by 300–400 degrees than for testing the Al-Co-Cr-Fe-Ni HEAs. On the other hand, Mn-Co-Cr-Fe-Ni HEAs have not been sufficiently studied, and the effect of additional elements has not been considered in practice; in particular, it can be assumed that the addition of aluminum would contribute to the formation of a continuous protective oxide film on the surface of these alloys.

## 4. High-Entropy Alloys from Refractory Metals

Refractory HEAs are alloys with several main components based on refractory elements such as W, Mo, Ta, Nb, V, Ti, Zr, and Hf [47,48]. Refractory HEAs crystallize with the formation of a bcc crystal lattice [49]. This group of HEAs has excellent mechanical properties at high temperatures, which is why the oxidative behavior of these alloys at elevated temperatures is also very important. It has been suggested recently [50,51,52] that HEA with high concentrations of refractory metals (usually exceeding 50 at. %) can be used as high-temperature structural materials, mainly due to their high melting points, retention of strength at elevated temperatures, sufficient plasticity and toughness at room temperature. However, the oxidation resistance of such HEAs is low, because refractory elements such as Ti, Zr and Hf have a strong affinity for oxygen, while their oxides have low adhesion to metal. In addition, oxides of V have a low melting point, while Mo- and W-oxides have low boiling points [8].

There are a number of studies of the behavior of refractory HEAs during high-temperature oxidation [49,53,54,55,56,57,58,59,60,61,62,63,64,65,66,67,68,69,70,71]. The most interesting results are given below; the role of aluminum in oxidation resistance should be noted.

Butler et al. [55] obtained NbTiZrV and NbTiZrCr alloys (without Al) of equimolar composition by vacuum arc melting, and studied the behavior of these alloys during oxidation in air at 1000 °C. The NbTiZrV alloy showed a high oxidation rate in air at 1000 °C and it was completely oxidized after 8 h. Oxidation in both alloys proceeded along grain boundaries by means of internal (diffusion) oxidation. The layer of the oxide consisted of TiO_2_, V_2_O_5_, TiNb_2_O_7_, and Nb_2_Zr_6_O_17_. Such a layer did not have sufficient protective properties.

Gorr et al. [58] obtained HEA AlCrMoNbTi of equimolar composition by arc melting, and investigated the oxidative behavior of this alloy in air at temperatures of 900 °C, 1000 °C and 1100 °C. The oxidation rate at 1000 °C was higher than at 900 °C; and the weight gain at 1000 °C was higher compared to 1100 °C. The surface of the alloy sample was covered with a thick, porous oxide layer that did not have protective properties, and consisted of various oxides. However, a discontinuous and thin layer rich in Cr oxide was formed in some places, which had some protective properties. In addition, the formation of an Al_2_O_3_ layer was observed at higher temperatures with prolonged exposure. The oxidation resistance of HEA improved significantly with the addition of 1 at. % Si. The total weight gain of the Si-containing HEA under the same experimental conditions was significantly lower than that of the HEA without Si. Thin and continuous oxide layers rich in Al and Cr formed on the surface of the Si-containing HEA. The authors explained the increase in oxidation resistance by the fact that Si additives affect the activity of Cr and/or Al in the alloy, leading to a higher driving force for the formation of protective layers of Cr_2_O_3_ and Al_2_O_3_.

Chang et al. [63] obtained a series of HfNbTaTiZr HEAs by vacuum arc melting, and studied the effect of the Al content on the oxidizing properties of samples of such alloys. A HEA without Al showed an increase in weight with an increase in the oxidation temperature. This clearly indicates that the oxide layers failed to provide high temperature oxidation protection. The addition of only 1 at. % Al increased oxidation resistance and provided a strong oxidation barrier in the 700–900 °C temperature range. However, the oxidation protection was not effective enough at the higher temperatures of 1100 °C and 1300 °C. The formation of less dense oxide layers at 1300 °C allowed O to easily diffuse into the HEA and thus contributed to a decrease in oxidation resistance.

Waseem and Ryu [69] developed a refractory HEA Al_x_Ta_y_V_z_Cr_20_Mo_20_Nb_20_Ti_20_Zr_10_ (Al_x_Ta_y_V_z_-Q). A high temperature oxidation study of Al_x_Ta_y_V_z_-Q alloy (carried out for 1 h at 1000 °C using thermogravimetric analysis) revealed the volatilization of oxides during alloy oxidation. The composition of the alloy was then modified to Al_10_Cr_y_Mo_x_Nb_y_Ti_y_Zr_10_ (Mo-x) in order to improve the oxidation resistance. Samples of this Mo-x alloy (where x is at. % Mo (Mo-0, Mo-4, Mo-8), y = (80 − x)/3)) were oxidized at 1000 °C for 50 h in air. Thermogravimetric analysis showed increased resistance to oxidation of the composition Al_10_CrNbTiZr_10_ (Mo-0). The increase in the sample mass after oxidation for 1 h at 1000 °C in air was only 1 mg/cm^2^. This was due to the formation of protective layers of Al and Cr oxides. Mo-4 and Mo-8 samples were completely oxidized after 10 h, while Al_10_CrNbTiZr_10_ (Mo-0) remained unchanged and showed an increase in mass of the order of 24 mg/cm^2^ only after 50 h. EDS analysis of the oxide film formed after 50 h of oxidation in the Mo-0 alloy revealed the presence of CrNbO_4_, Al_2_O_3_, and AlTiO_5_ in its composition, which explained the increased resistance to oxidation. The results showed the potential of this alloy for high-temperature applications.

Müller et al. [70] studied the mechanism of high-temperature oxidation of a number of refractory HEAs: AlCrMoNbTi (Figure 10), AlCrMoNb (Figure 11), AlCrMoTaTi (Figure 12) and AlCrMoTa (Figure 13) in the temperature range from 900 °C to 1100 °C in air. The authors sought to elucidate the role of the elements Ti, Nb and Ta on the resistance to oxidation. The studies showed the excellent resistance of these alloys to oxidation in air. The authors associate this with the formation of protective layers of the oxides Al_2_O_3_, Cr_2_O_3_ and CrTaO_4_ (in particular, for AlCrMoTaTi at 1000 °C), and with the slow diffusion of oxygen through the CrTaO_4_ layer. Although in systems with Nb (AlCrMoNbTi and AlCrMoNb), protective oxide layers consisting of Al_2_O_3_, Cr_2_O_3_, and CrNbO_4_ also formed, the strong anisotropic thermal expansion of polymorphic modifications of Nb_2_O_5_ led to the formation of pores and the exfoliation of the oxide layer. The presence of Ti in the AlCrMoNbTi and AlCrMoTaTi alloys played a decisive role in the formation of protective layers of rutile-type oxides (such as CrTaO_4_), while simultaneously reducing the number of oxides that negatively affect the oxidation resistance (Nb_2_O_5_, Ta_2_O_5_). The study, according to the authors, opens the way for the further development of oxidation-resistant refractory HEAs.

Cao et al. [71] studied high-temperature oxidation (at 1000 °C) of three alloys: NbTa_0.5_TiZr, AlNbTa_0.5_TiZr, and AlMo_0.5_NbTa_0.5_TiZr. The results showed that the rates of oxidation of alloys NbTa_0.5_TiZr (Figure 14) and AlNbTa_0.5_TiZr (Figure 15) are determined by diffusion and these alloys are oxidized exponentially. However, the oxidation rate of the AlMo_0.5_NbTa_0.5_TiZr alloy (Figure 16) is limited by the interphase reaction and oxidation and linearly.

The law describing the oxidation process (linear oxidation or exponential oxidation [49,72]) makes it possible to judge the stability of oxidizing alloys. Linear oxidation indicates the low resistance of the alloy due to the fact that the oxide layers weakly protect against further oxidation. Exponential oxidation usually indicates high oxidation stability (due to the formation of protective oxide layers and low oxygen solubility).

Cao et al. [71] determined that TiO_2_, Nb_2_O_5_, Ti_3_O_5_, and ZrO_2_ are the main components of the oxide layer on the NbTa_0.5_TiZr alloy sample. The presence of Al in the alloy led to the appearance, in the oxide layer on the sample of the AlNbTa_0.5_TiZr alloy, of a dense layer of Al_2_O_3_ with high adhesion to the base. A protective layer of Al_2_O_3_ ensured the high resistance to oxidation of refractory HEAs. The addition of Mo caused a significant decrease in oxidation resistance, mainly due to the formation of MoO_3_, which evaporated during high temperature oxidation. Evaporating MoO_3_ left extensive pores in the subsurface zone and promoted the formation of cracks, which led to rapid oxidation of the alloy AlMo_0.5_NbTa_0.5_TiZr.

The data from different authors on the weight gain during the oxidation of HEAs of refractory metal systems are summarized in Table 4. The data are arranged in descending order of weight gain (in bold) for 1 h at 1000 °C.

According to the data from Table 4, the best corrosion resistance was observed for the alloys HfNbTiZr [61], AlCrMoTaTi [66] and AlTaMoCr [70]. Moreover, as noted by researchers [66,70], a combination of molybdenum and tantalum makes it possible to obtain such excellent properties. All three alloys have a varied composition of the oxide film. For the HfNbTiZr alloy, the oxidation products contain NbO, ZrO_2_, TiO_2_; for the AlCrMoTaTi alloy—Al_2_O_3_, Cr_2_O_3_, TiO_2_, CrTaO_4_, for the AlTaMoCr alloy—Cr_2_O_3_, Al_2_O_3_, CrTaO_4_. Note that, for the last two compositions, Mo compounds are absent in the oxidation products.

A variety of compositions of refractory HEAs can be noted, which speaks more of a certain haphazard development of compositions than a logically built chain of experiments. It can also be immediately noted that HEAs, consisting entirely of only refractory metals, have a relatively low resistance to high-temperature gas corrosion. The situation changes as soon as aluminum and/or chromium are additionally introduced into the composition. Thus, it seems that the creation of alloys based on the basic composition Al-Co-Cr-Fe-Ni with the addition of refractory elements (for example, the combination of tantalum with molybdenum, hafnium or zirconium) is perspective trend for investigators.

The summary information given in Table 4 is supplemented with the activation energy calculated by us (data used for holding for 20 h). The obtained values of the activation energy correlate well with the experimental data on the change in the mass of the samples - materials with a high activation energy have a high resistance to oxidation.

## 5. HEAs Oxidation Products

Our analysis of the literature made it possible to compile a list of HEA oxidation products found in experiments (Table 5). The data are presented as the composition of the formed oxide film becomes more complex. Šulhánek P. et al. [73] were found that for the formation of a protective film consisting only of Al_2_O_3_, the required concentration of aluminum in the alloy must be more than 20 at. %. For the presented in Table 5 alloys, the Al concentration does not reach this indicator; therefore, the composition of the observed oxide films is very complex and varied. Generalizations from such information is important not only from the point of view of predicting the possible protective properties of the scale formed, but also for analyzing the possibility of obtaining metal-strengthening ceramic particles during the planned saturation of the surface layers of the alloy with O.

Such information is of great value as experimental data for comparison with the results of the models of the oxidation process. The significant amount of empirical data accumulated to date can allow deeper theoretical generalizations. One of these approaches is the thermodynamic modeling of phase equilibria, which is known as a powerful tool for the analysis of high-temperature processes, including high-temperature oxidation. It can make it possible to predict the composition of oxidation products of complex metal alloys, depending on their compositions and the oxidation conditions.

However, if we talk about the thermodynamic characteristics of high-entropy systems, at present thermodynamic calculations are reduced mainly to calculating the entropy and enthalpy of mixing in multicomponent systems [23]. There are other works which carry out the modeling of phase equilibria, which are realized during the crystallization of HEAs, using CALPHAD algorithms. Diagrams showing thermodynamically calculated phase diagrams can be found in [29,74,75,76,77].

There are also works where the results of calculating phase diagrams are presented in the form of isocomposite sections. Zhang C. et al. [76] discuss the Al_x_-Co-Cr-Fe-Ni system. Zhang F. et al. [78] present the results of calculating the phase diagrams of the Co-Cr-Fe-Mn-Ni, Co-Cr-Fe-Mn-Cu and Cr-Nb-Ti-V-Zr systems. Guruvidyathri et al. [79] present the results of modeling phase equilibria for 52 HEAs, comparing them with experimental results. The data presented in [79] show that thermodynamic modeling can be successfully used to predict the formation of undesirable intermetallic phases (in particular, topologically close-packed phases), especially for HEAs containing transition metals.

Modeling the results of processes occurring during high-temperature oxidation of HEAs is extremely rare in the literature. Ferrari and Körmann [80] attempted to calculate the segregation energy of atoms in the MnCoCrFeNi alloy in a vacuum and at atmospheric pressure for interaction with oxygen atoms. Osei-Agyemang and Balasubramanian [81] showed the calculation of the change in the Gibbs free energy of adsorption of oxygen atoms on the HEAs surface depending on the partial pressure of oxygen from 10^–9^ to 10^2^ bar for 300–1500 K.

An example of the use of CALPHAD for modeling phase equilibria during high-temperature oxidation of HEAs is found in [82] for Al_x_-Co-Cr-Fe-Ni (x = 8, 12, 15, 20, 30 at. %) based HEAs. The example of results of modeling the high-temperature oxidation of the samples under study using the CALPHAD algorithms is shown in Figure 17. The calculation was performed using the TCNI8 Ni-based superalloys database in ThermoCalc. In these calculations Butler used the modeling method for the Co-Al-W system in [83].

According to the simulation results [82] (see Figure 17), oxide phases begin to form on the surface of the Al_x_CoCrFeNi alloy when oxygen activity in the gas phase is more than 10^–14^. At sufficiently high values of oxygen activity (close to unity), spinels should act as oxidation products, according to the modeling. With oxygen activity in the gas phase in the range 10^–8^–10^–14^ on the surface of the samples, according to calculations, a solid solution of oxides (Al,Cr)_2_O_3_ should form, excluding the Al_15_ sample, where only corundum is formed.

Butler [82] concludes that the results of modeling correspond to the results of his experiments, and a similar thermodynamic analysis can be used to predict the behavior of HEAs at high temperatures. However, work on this modeling for other HEAs has not been carried out, which opens up great prospects in terms of modeling phase equilibria during gas corrosion.

## 6. Conclusions

This review and analysis of data on the oxidative behavior of HEAs leads to the following conclusions.(1)Promising heat-resistant and corrosion-resistant HEAs can be found in alloys of the Al-Co-Cr-Fe-Ni system. The addition of other elements to this base alloy has a different effect on its properties: additions of Si, Ti, Y and Hf will increase the resistance of such alloys to high-temperature gas corrosion; Mo additions have a negative effect on corrosion resistance, since Mo oxides are characterized by low evaporation temperatures a and even at low concentrations of Mo, the presence of Mo oxides in the oxide layer can lead to a violation of the integrity of the scale and make it loose and porous; the introduction of Zr and Ta is not fully understood; the effect of Cu additions also requires additional research, but it is already obvious that at sufficiently high Cu concentrations (when its amount is comparable to the amounts of other elements forming a multicomponent base), the adhesion of the formed oxide layer with respect to the matrix surface is low; such scale peels off and breaks easily.(2)HEAs of the Mn-Co-Cr-Fe-Ni system are less resistant to high-temperature gas corrosion than alloys without Mn. The mechanism of oxidation of such alloys still needs to be studied; however, it is known that a multiphase, discontinuous scale is formed on such alloys, while the scale layers containing Mn oxide are porous, peel off easily, and do not interfere with the further oxidation of the alloy.(3)Studies of the resistance to high-temperature corrosion of HEAs based on refractory metals are contradictory. According to the literature, the alloys most resistant to high-temperature oxidation are the HEAs of the Al-Cr-Mo-Ta-Ti system. In the presence of Ta, Mo does not participate in the oxidation process and, therefore, there is no Mo oxide in the scale. In the absence of Al, oxides of refractory metals do not have sufficient protective properties.(4)The most important elements that contribute to an increase in the resistance of HEAs to oxidation are Al and Cr. Continuous films of Al_2_O_3_ or Cr_2_O_3_ with high adhesion to the alloy can be formed, which prevent the directional diffusion of oxygen into the alloy matrix. These same elements can react with other elements in the HEA (for example, Fe and Ni), which can lead to the formation of spinel films, which also increase resistance to oxidation. Elements such as Si and Y can also improve oxidation resistance.(5)The presence of refractory elements, such as W, Mo, Ta, Nb, V, Ti, Zr and Hf, can increase the strength of HEAs at high temperatures, but some of these elements (primarily Mo, W, and V) can accelerate the oxidation of alloys, primarily due to the formation of volatile oxides.(6)When Ti is one of the main elements of an alloy, it plays an important role in the formation of protective layers from CrTaO_4_, TiO_2_, Ti_3_O_5_ and ZrO_2_ oxides, while simultaneously reducing the number of oxides that reduce heat resistance (Nb_2_O_5_, Ta_2_O_5_).(7)The creation of alloys based on the basic composition Al-Co-Cr-Fe-Ni with the addition of refractory elements (for example, the combination of tantalum with molybdenum, hafnium or zirconium) is perspective trend for investigators.(8)Several gaps can be noted in the study of high-temperature oxidation of HEAs:(8.1)Thermodynamic modeling tools (for example, CALPHAD) are practically not used to predict high-temperature oxidation processes. The use of thermodynamic modeling techniques would optimize the search for compositions of corrosion-resistant HEAs.(8.2)All studies of high-temperature oxidation of HEAs are empirical. In the literature, there are no theoretical data on the mechanism of the formation of protective films on HEAs and the effect of additional elements on such mechanisms.(8.3)For the study of high-temperature gas corrosion HEAs, the authors of the presented articles proposed only one method - oxidation in air. However, for practical use, it would be beneficial to conduct tests in atmospheres of H_2_/H_2_O (water vapor), SO_2_/SO_3_ (typical for the oil and gas industry) or CO/CO_2_ (when engines are running in aircraft and rocketry mechanisms).

## Figures and Tables

**Figure 1 materials-14-02595-f001:**
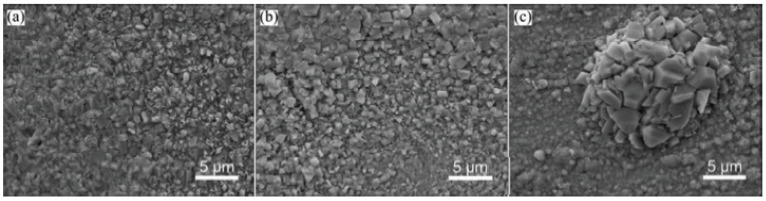
Surface morphologies of oxide films formed on the surfaces of alloys after 70 h oxidation at 600 °C under 23 MPa [27]: (**a**) Al_0.15_; (**b**) Al_0.4_; (**c**) HR3C.

**Figure 2 materials-14-02595-f002:**
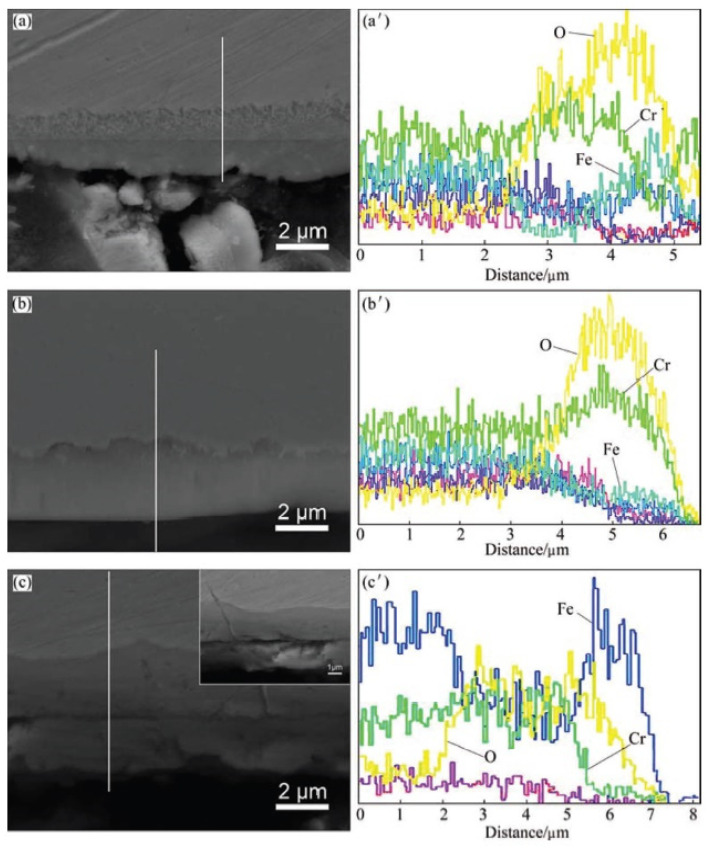
SEM images (**a**–**c**) and EDS line scans (**a’**–**c’**) at cross-section of surfaces of alloys after 70 h oxidation at 600 °C at 23 MPa [27]: (a, a’) Al_0.15_; (b, b’) Al_0.4_; (c, c’) HR3C.

**Figure 3 materials-14-02595-f003:**
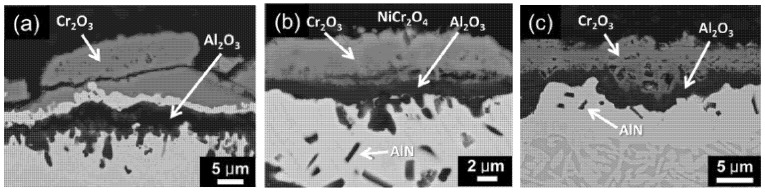
Cross-sectional electron images of the HEAs after 50 h oxidation at 1050 °C in air: (**a**) Al_8_; (**b**) Al_12_; (**c**) Al_20_ [29].

**Figure 4 materials-14-02595-f004:**
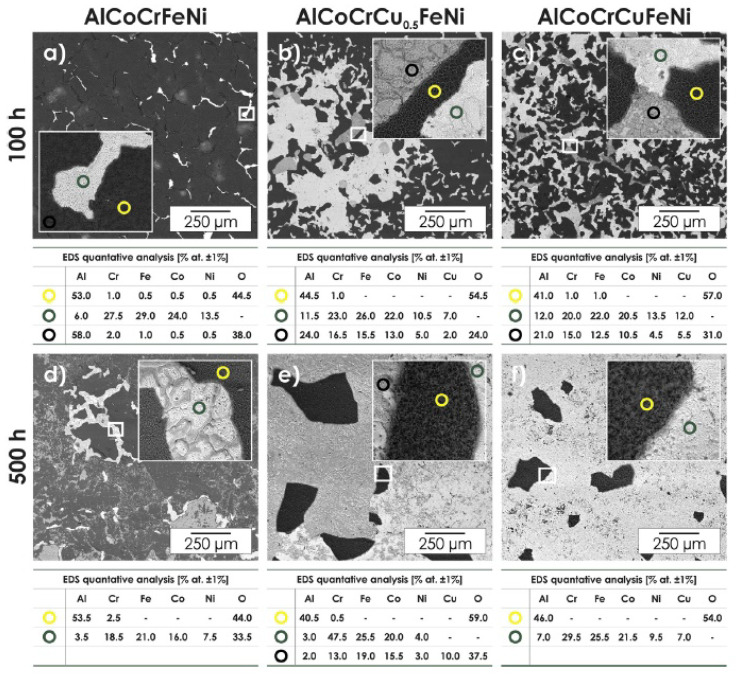
The back-scattered electrons (BSE) microphotography’s of the samples’ morphology after 100 h of oxidation at 1000 °C (**a**–**c**) and after 500 h of oxidation at 1000 °C (**d**–**f**) [32]: (**a**,**d**) AlCoCrFeNi; (**b**,**e**) AlCoCrCu_0.5_FeNi; (**c**,**f**) AlCoCrCuFeNi. The circles in the images indicate the places of the EDS chemical analysis for each alloy, the analysis results are presented in the tables.

**Figure 5 materials-14-02595-f005:**
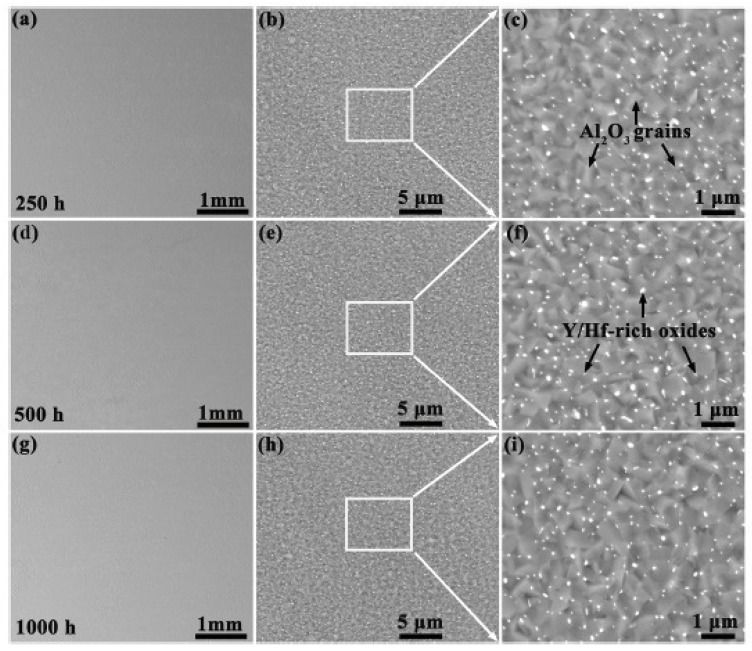
The morphology of the surface layer of scale formed on the AlCoCrFeNiYHf alloy after isothermal oxidation at 1100 °C: (**a**–**c**) 250 h; (**d**–**f**) 500 h and (**g**–**i**) 1000 h [28].

**Figure 6 materials-14-02595-f006:**
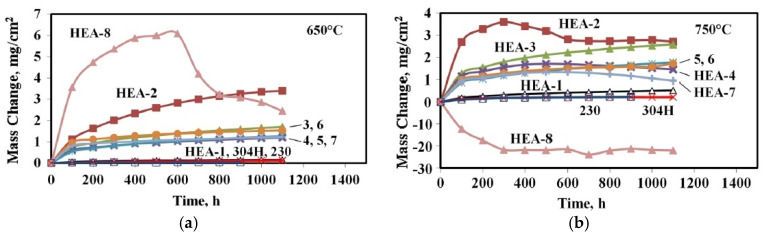
Weight change in air measurements versus time: (**a**) at 650 °C: (**b**) at 750 °C [42].

**Figure 7 materials-14-02595-f007:**
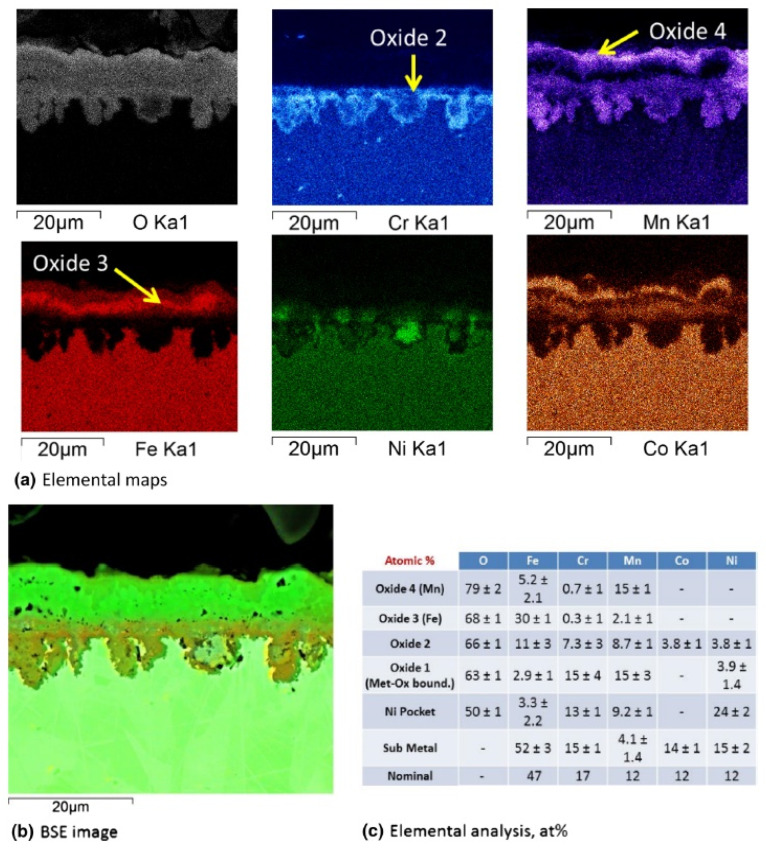
HEA-7 exposed for 1100 h at 650 °C in air [42]: (**a**) elemental maps; (**b**) BSE image; (**c**) elemental analysis, at. %.

**Figure 8 materials-14-02595-f008:**
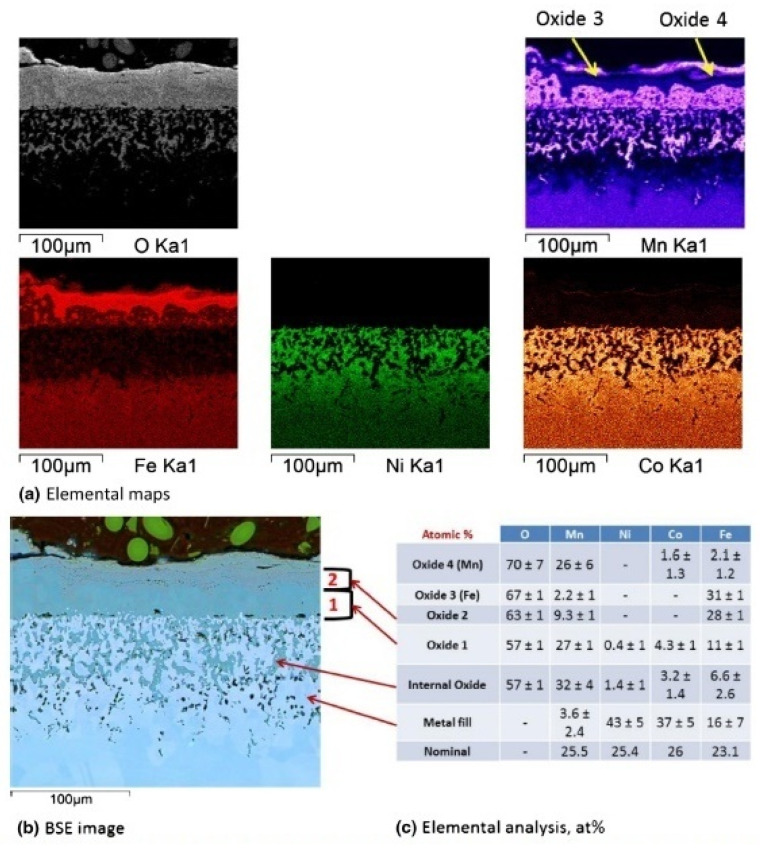
HEA-8 exposed for 1100 h at 650 °C in air [42]: (**a**) elemental maps; (**b**) BSE image; (**c**) elemental analysis, at. %.

**Figure 9 materials-14-02595-f009:**
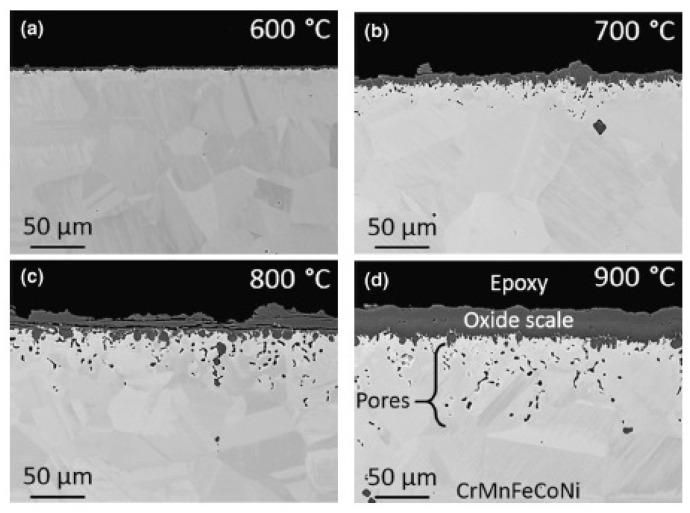
SEM images of oxide layers on HEA MnCoCrFeNi in cross section after 100 h at: (**a**) 600 °C; (**b**) 700 °C; (**c**) 800 °C; (**d**) 900 °C [43].

**Figure 10 materials-14-02595-f010:**
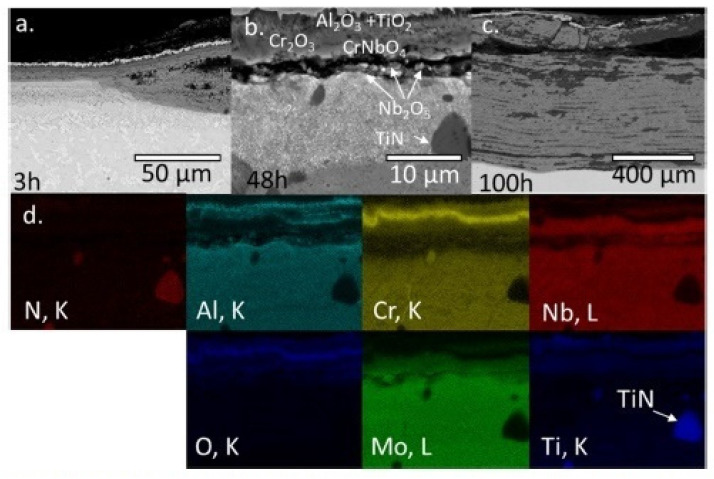
BSE images of AlCrMoNbTi after 3 h (**a**), 48 h (**b**) and 100 h (**c**) of exposure to air at 1000 °C and (**d**) the corresponding EDX mapping of (**b**) [70].

**Figure 11 materials-14-02595-f011:**
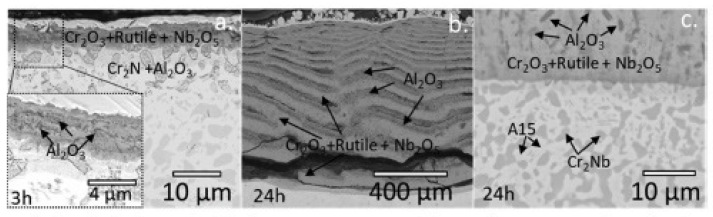
SEM images of AlCrMoNb after 3 h (**a**), 48 h (**b**) and 100 h (**c**) at 1000 °C in air. Phase A15 in the photo (c)—Al(Nb, Mo)_3_ [70].

**Figure 12 materials-14-02595-f012:**
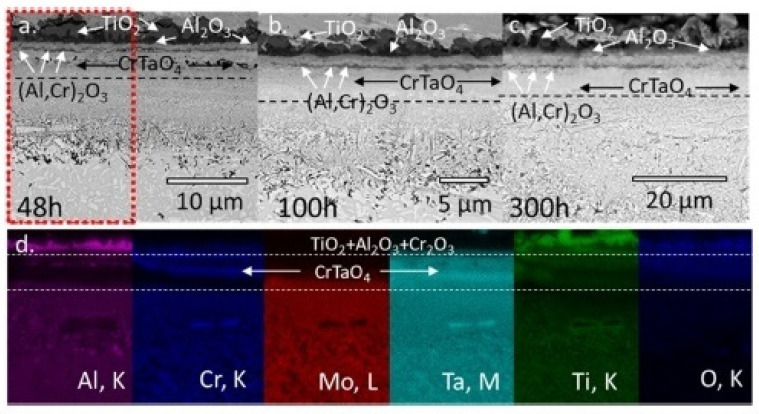
SEM images of AlCrMoTaTi after oxidation in air: after 48 h (**a**), 100 h (**b**) and 300 h (**c**) at 1000 °C and mapping EDS (**d**) in cross section (marked area in (a)) [70].

**Figure 13 materials-14-02595-f013:**
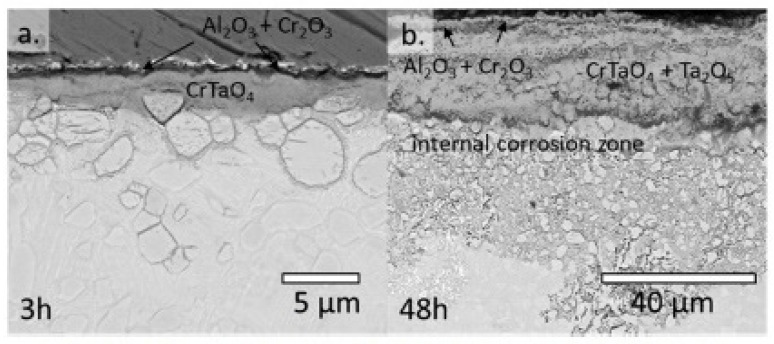
SEM images of AlCrMoTa after oxidation in air: after 3 h (**a**), 48 h (**b**) at 1000 °C [70].

**Figure 14 materials-14-02595-f014:**
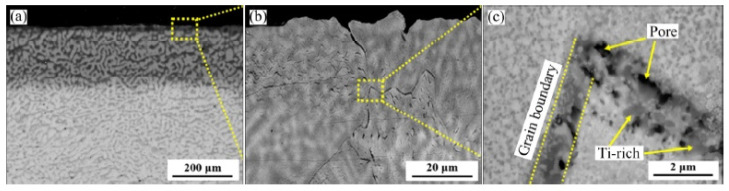
Microstructures of oxide scale of NbTa_0.5_TiZr alloy oxidized at 1000 °C for 10 h: (**a**) general view; (**b**), (**c**) enlarged fragments [71].

**Figure 15 materials-14-02595-f015:**
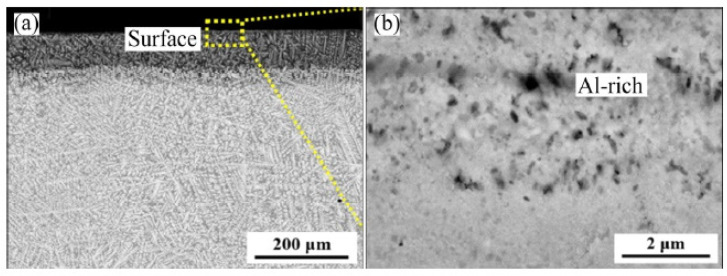
Microstructures of oxide scales of AlNbTa_0.5_TiZr alloy oxidized at 1000 °C for 10 h: (**a**) general view; (**b**) details of oxides in surface [71].

**Figure 16 materials-14-02595-f016:**
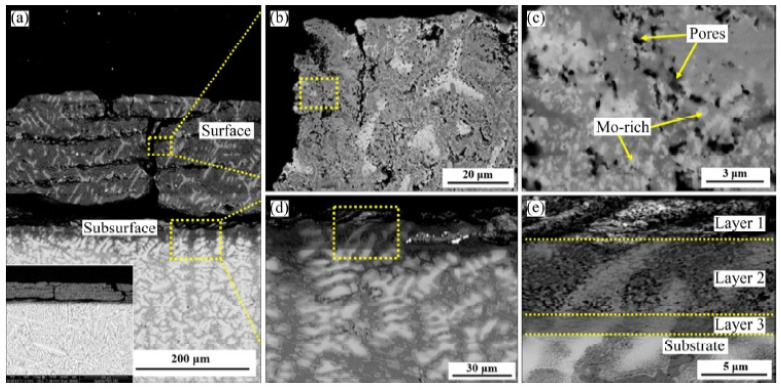
Microstructures of oxide scales of AlMo_0.5_NbTa_0.5_TiZr alloy oxidized at 1000 °C for 10 h: (**a**) general view; (**b**–**e**) enlarged fragments [71].

**Figure 17 materials-14-02595-f017:**
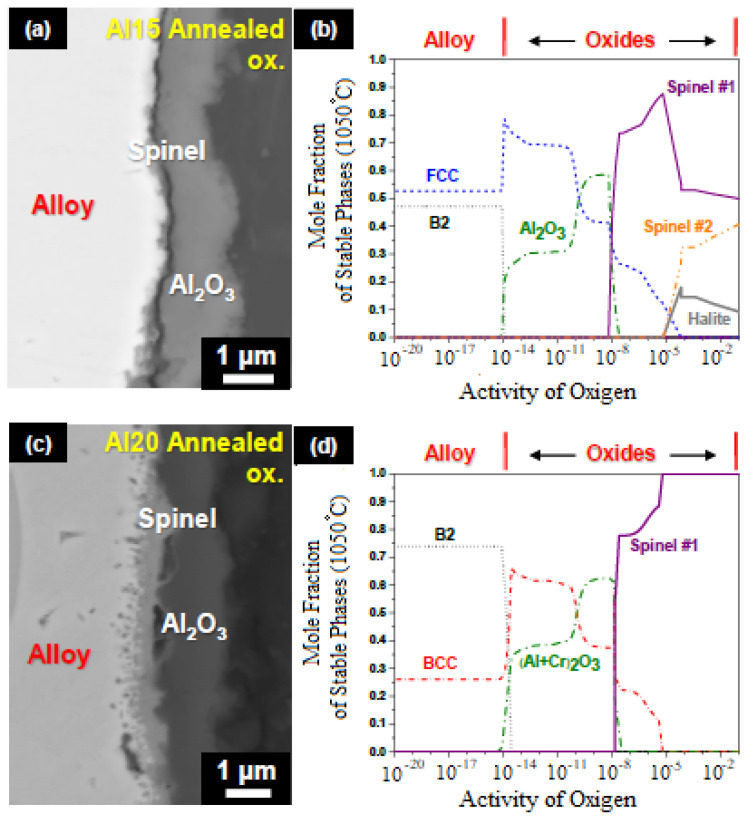
90° Clockwise rotated BSE images of the annealed Al_15_ and Al_20_ HEAs after 50 h of oxidation (**a**,**c**), respectively; along with thermodynamically calculated 1050 °C isothermal phase diagrams for the Al_15_ and Al_20_ HEAs with varying oxygen activities (**b**,**d**), respectively [82].

**Table 1 materials-14-02595-t001:** Weight gain during oxidation in air of HEAs based on the Al-Co-Cr-Fe-Ni system.

Alloy	t, °C	Weight Gain (mg/cm^2^)				Source	Activation Energy, kJ/mol
		1 h	5 h	20 h	50 h		
In descending order weight gain for 1 h							
Al_15_(CoCrFeNi)_85_	1050	0.40	0.85	1.62	1.68	[29]	–
Al_20_(NiCoCrFe)_80_ (annealed)	1050	0.20	0.81	1.16	1.52	[30]	–
Al_8_(CoCrFeNi)_92_	1050	0.18	0.82	1.50	1.70	[29]	–
Al_10_(CoCrFeNi)_90_	1050	0.17	0.72	1.28	1.40	[29]	–
Al_12_(CoCrFeNi)_88_	1050	0.15	0.63	1.08	1.12	[29]	–
Al_20_(CoCrFeNi)_80_	1050	0.11	0.62	0.88	0.98	[29]	–
Al_15_(CoCrFeNi)_85_ (annealed)	1050	0.10	0.90	1.40	1.83	[30]	–
Al_8_(CoCrFeNi)_92_ (annealed)	1050	0.10	0.61	1.30	1.85	[30]	–
Al_12_(CoCrFeNi)_88_ (annealed)	1050	0.10	0.60	1.17	1.60	[30]	–
Al_30_(CoCrFeNi)_70_	1050	0.07	0.45	0.50	0.49	[29]	–
Al_4_Co_5_Cr_5_Ni_5_Si	1050	0.06	0.35	0.40	0.43	[31]	–
Al_2_Co_4.5_Cr_4.5_Ni_4.5_Fe_4.5_	1050	0.05	0.40	0.85	1.50	[31]	–
Al_3_Co_2_Cr_7_Ni_7_Si	1050	0.04	0.09	0.08	0.09	[31]	–
Al_30_(CoCrFeNi)_70_ (annealed)	1050	0.03	0.45	0.52	0.60	[30]	–
In descending order weight gain for 20 h at 1000 °C							
Al_0.5_CoCrCu_0.5_FeNi_2_	800	–	–	4.00	18.00	[33]	199
	1000	–	–	23.00	40.00		
Al_1.5_CoCr_1.5_Cu_0.5_FeNi	800	–	–	3.00	19.00	[33]	137
	1000	–	–	10.00	17.00		
AlCoCrCuFeNi	800	–	–	3.00	11.00	[33]	125
	1000	–	–	9.00	10.00		
AlCoCrFeNi	1000	–	0.27	0.50	0.72	[32]	–
AlCoCrCu_0.5_FeNi	1000	–	0.21	0.32	0.43	[32]	–
AlCoCrCuFeNi	1000	–	0.02	0.11	0.21	[32]	–
In descending order weight gain for 1 h at 1100 °C							
AlCoCrFeMo_0.5_NiSiTi	500	0.9	1.2	1.7	2.3	[25]	35
	800	0.3	1	2.2	3.7		
	1100	2.5	3.5	6.2	9		
AlCrFeMo_0.5_NiSiTi	500	1.3	1.7	2.8	3	[25]	29
	800	1	1.8	2.8	8		
	1100	2.2	3.1	4.9	7.1		

**Table 2 materials-14-02595-t002:** Chemical composition of the alloys [42].

Alloy	Fe	Ni	Co	Mn	Cr	Nb	Al	other	Y	O	N	C	S
	weight %								ppm				
HEA-1	24.85	25.89	26.00	0.51	22.66	–	0.07	–	–	14	67	82	135
HEA-2	17.39	21.47	21.68	20.27	19.14	–	–	–	–	11	64	68	136
HEA-3	17.18	22.15	21.45	19.91	19.17	0.11	–	–	1206	19	19	219	14
HEA-4	31.01	17.86	17.63	16.25	17.15	0.09	–	–	418	14	51	174	4
HEA-5	29.26	17.57	16.95	15.24	20.85	0.09	–	–	1033	3	15	159	11
HEA-6	39.80	15.42	14.89	13.62	16.06	0.17	–	–	969	10	7	262	<1
HEA-7	46.86	12.88	12.54	11.72	15.80	0.16	–	–	357	5	29	303	15
HEA-8	22.63	26.06	26.82	24.47	–	–	–	–	243	5	10	81	26
304H	70.51	8.29	0.23	1.09	18.71	–	0.09	0.46 Si	–	–	–	400	40
230	0.40	59.76	0.33	0.51	22.15	–	0.46	14.40 W	–	–	–	1000	–
								1.26 Mo					
								0.47 Si					

**Table 3 materials-14-02595-t003:** Weight gain during oxidation in air of HEAs based on the Mn-Co-Cr-Fe-Ni system.

Alloy	t, °C	Weight Gain (mg/cm^2^)				Activation Energy, kJ/mol	Source
		5 h	20 h	50 h	100 h		
Mn_24_Co_26_Fe_23_Ni_26_ (with addition of Y) (HEA-8)	650	–	–	20	3.5	–	[42] In descending order weight gain for 1 h at 650 °C
	750	–	–	−6	−13		
Mn_20_Co_22_Cr_19_Fe_17_Ni_21_ (HEA-2)	650	–	–	0.5	1.2	111	
	750	–	–	1.5	2.8		
Mn_20_Co_22_Cr_19_Fe_17_Ni_22_ (with addition of Y) (HEA-3)	650	–	–	0.5	1.0	28	
	750	–	–	0.8	1.3		
Mn_14_Co_15_Cr_16_Fe_40_Ni_15_ (with addition of Y) (HEA-6)	650	–	–	0.5	0.9	77	
	750	–	–	0.5	1.0		
Mn_16_Co_18_Cr_17_Fe_31_Ni_17_ (with addition of Y) (HEA-4)	650	–	–	0.4	0.9	137	
	750	–	–	0.7	1.1		
Mn_15_Co_17_Cr_21_Fe_29_Ni_17_ (with addition of Y) (HEA-5)	650	–	–	0.4	0.5	56	
	750	–	–	0.6	1.0		
Mn_12_Co_12_Cr_16_Fe_47_Ni_13_ (with addition of Y) (HEA-7)	650	–	–	0.4	0.5	123	
	750	–	–	0.5	1.0		
Mn_0.5_Co_26_Cr_22_Fe_25_Ni_26_ (HEA-1)	650	–	–	0.10	0.15	162	
	750	–	–	0.05	0.10		
MnCoCrFeNi	600	–	0.10	0.20	0.35	130	[43]
	700	0.10	0.35	0.65	1.05		
	800	0.25	0.75	1.40	2.25		
	900	0.60	1.25	2.15	–		

**Table 4 materials-14-02595-t004:** Weight gain during oxidation of refractory HEAs in air.

Alloy	t, °C	Weight Gain (mg/cm^2^)				Source	Activation Energy, kJ/mol
		1 h	3 h	20 h	50 h		
CrMo_0.5_NbTa_0.5_TiZr	1000	55	110	–	–	[49]	–
NbTiVZr	1000	30	100	–	–	[55]	–
HfNbTaTiZr	700	3	9	45	53	[63]	59
	900	18	25	29	39		
	1000	17	–	–	–		
	1100	20.5	30	48	51		
	1300	25	75	225	250		
NbTa_0.5_TiZr	1000	15	35	90	–	[71]	–
Al_0.5_Cr_0.5_MoNbTiZr	1000	13.38	–	–	–	[64]	–
NbTiZrCr	1000	13	25	60	83	[55]	–
AlCrNbTiZr	1000	10.66	–	–	–	[64]	–
Cr_1.5_Mo_0.5_NbTiZr	1000	9.4	–	39	–	[64]	–
CrMoNbTiZr	1000	8.77	–	–	–	[64]	–
Al_0.5_HfNbTaTiZr	700	3	4	8	10	[63]	132
	900	6	10	11	12		
	1000	8.5	–	–	–		
	1100	12	17	25	38		
	1300	25	100	248	250		
AlMoNbTiZr	1000	8.25	–	–	–	[64]	–
Al_1.5_Mo_0.5_NbTiZr	1000	7.29	–	–	–	[64]	–
Al_1.5_Cr_0.5_NbTiZr	1000	6.3	–	20	–	[64]	–
Al_0.5_Cr_1.5_NbTiZr	1000	6.1	–	–	–	[64]	–
AlHfNbTaTiZr	700	1	2	5	8	[63]	137
	900	5	6	9	11		
	1000	6	–	–	–		
	1100	8	14.5	18.5	33		
	1300	25	49	177	250		
AlMo_0.5_NbTa_0.5_TiZr	1000	5	17	75	–	[71]	–
AlNb_1.5_Ta_0.5_Ti_1.5_Zr_0.5_	1000	5	10	25	38	[68]	–
AlCr_0.5_Mo_0.5_NbTiZr	1000	4.29	–	21	–	[64]	–
AlCrNbTiZr	800	3	5	8	10	[65]	180
	1000	4	13	23	52		
	1200	5	17	90	185		
AlNbTa_0.5_TiZr	1000	4	10	35	–	[71]	–
AlNbTiZr	1000	3.8	–	–	–	[61]	–
Al_0.5_CrMo_0.5_NbTiZr	1000	3.46	–	–	–	[64]	–
CrMoNbTaV	900	2	8	25	–	[67]	92
	1000	3	13	42	–		
	1100	7.5	22	12	–		
Al_10_Cr_24_Mo_8_Nb_24_Ti_24_Zr_10_ (Mo-8)	1000	3.0	–	–	–	[69]	–
AlCrMoTiW	1000	2.3	3.8	6.2	–	[54]	–
Al_10_Cr_25_Mo_4_Nb_25_Ti_25_Zr_10_ (Mo-4)	1000	2.0	–	–	–	[69]	–
Al_0.5_Mo_1.5_NbTiZr	1000	1.27	–	–	–	[64]	–
AlCrMoTaTi	1000	1.15	1.25	1.9	2.2	[70]	–
Al_10_Cr_27_Nb_27_Ti_27_Zr_10_ (Mo-0)	1000	1.0	–	17	24	[69]	–
AlCrMoNbTi-1at%Si	900	0.1	0.2	0.3	0.5	[58]	309
	1000	0.5	1	2.2	5.9		
	1100	0.6	1.2	2.9	6.4		
AlCrMoNbTi	900	0.1	0.2	0.3	0.4	[58]	395
	1000	0.5	0.7	3.2	9		
	1100	0.7	0.9	5.5	8		
Cr_0.5_Mo_1.5_NbTiZr	1000	0.5	–	–	–	[64]	–
AlNbMoCr	1000	0.3	0.6	14.3	–	[70]	–
AlCrMoNbTi	1000	0.3	0.6	3.1	12.5	[70]	–
AlCrMoTaTi-1at%Si	900	0.1	0.15	0.35	0.45	[66]	134
	1000	0.25	0.4	0.8	1.3		
	1100	1	1.55	2.6	4		
HfNbTiZr	1000	0.25	–	–	–	[61]	–
AlCrMoTaTi	900	0.1	0.15	0.2	0.25	[66]	284
	1000	0.15	0.3	0.5	0.6		
	1100	0.55	1	2	3.05		
AlTaMoCr	1000	~0	0.3	0.4	1.3	[70]	–

**Table 5 materials-14-02595-t005:** Composition of HEAs oxidation products.

Alloy	t, °C	Time, h	Oxidation Products	Source
AlCrMoTiW	1000	40	Cr_2_O_3_, Al_2_O_3_	[54]
AlCrMoNbTi	900–1100	48	Cr_2_O_3_, Al_2_O_3_	[58]
NbTiZrCr	1000	100	NbCrO_4_, ZrO_2_	[55]
Al_2_Co_4.5_Cr_4.5_Fe_4.5_Ni_4.5_	1050	5–500	Cr_2_O_3_, Al_2_O_3_, AlN	[31]
Al_4_Co_5_Cr_5_Ni_5_Si	1050	5–500	Cr_2_O_3_, Al_2_O_3_, AlN	[31]
Al_3_Co_7_Cr_2_Ni_7_Si	1050	5–500	Cr_2_O_3_, Al_2_O_3_, AlN	[31]
AlCoCrCu_x_FeNi (x = 0, 0.5 and 1.0 at.%)	1000	100 500	Al_2_O_3_, spinels with Cr and Co	[32]
Al_0.5_CoCrCu_0.5_FeNi_2_	800–1000	200	Cr_2_O_3_, Al_2_O_3_, oxides of Fe and Ni	[33]
Al_1.5_CoCr_1.5_Cu_0.5_FeNi	800–1000	200	Cr_2_O_3_, Al_2_O_3_, oxides of Fe and Ni	[33]
AlCoCrCuFeNi	800–1000	200	Cr_2_O_3_, Al_2_O_3_, oxides of Fe and Ni	[33]
Al_x_(CoCrFeNi) (x = 8, 10, 12, 15, 20 and 30 at.%)	1050	100	Cr_2_O_3_, Al_2_O_3_, NiCr_2_O_4_, AlN	[29]
AlCoCrFeNi-0.02at%Y-0.02at%Hf	1100	1–1000	Al_2_O_3_, Y_3_Al_5_O_12_, HfO_2_	[28]
MnCoCrFeNi (HEA 1-HEA 8)	650–750	1100	Cr_2_O_3_, oxides of Fe and Mn, MnCr_2_O_4_	[42]
CoCrFeMnNi	500–900	100	α-Mn_2_O_3_, Mn_3_O_4_, Cr_2_O_3_	[43]
AlCrMoTaTi	900–1100	3–300	Cr_2_O_3_, Al_2_O_3_, CrTaO_4_	[70]
AlCrMoTa	900–1100	3–300	Cr_2_O_3_, Al_2_O_3_, CrTaO_4_	[70]
AlCrMoNbTi	900–1100	3–300	Cr_2_O_3_, Al_2_O_3_, Nb_2_O_5_	[70]
AlCrMoNb	900–1100	3–300	Cr_2_O_3_, Al_2_O_3_, Nb_2_O_5_	[70]
AlCrMoNbTi	1000–1100	48	Al_2_O_3_, Cr_2_O_3_, TiO_2_	[60]
AlCrMoTiW	1000–1100	48	Al_2_(WO_4_)_3_, Cr_2_O_3_, TiO_2_	[59]
Al_x_CoCrFeNi (x = 0.15, 0.4 at.%)	550–600	70	Fe_3_O_4_, FeCr_2_O_4_, NiFe_2_O_4_ (Cr_2_O_3_) (Al_2_O_3_)	[27]
AlCrMoTaTi	900–1100	48	Al_2_O_3_, Cr_2_O_3_, TiO_2_, CrTaO_4_	[66]
AlCrMoTaTi-1at%Si	900–1100	48	Al_2_O_3_, Cr_2_O_3_, TiO_2_, CrTaO_4_	[66]
AlCrMoTaTi	1000–1100	48	Al_2_O_3_, Cr_2_O_3_, TiO_2,_ CrTaO_4_	[60]
NbTa_0.5_TiZr	1000	10	TiO_2_, Nb_2_O_5_, Ti_3_O_5_, ZrO_2_	[71]
CrNbTi-10at%Al-10at%Zr-x at%Mo	1000	1–50	Cr_2_O_3_, Al_2_O_3_, AlTiO_5_, CrNbO_4_	[69]
NbTiVZr	1000	100	TiO_2_, V_2_O_5_, TiNb_2_O_7_, Nb_2_Zr_6_O_17_	[55]
AlNbTa_0.5_TiZr	1000	10	TiO_2_, Nb_2_O_5_, Ti_3_O_5_, ZrO_2_, Al_2_O_3_	[71]
AlCrNbTiZr	800–1200	5–50	CrNbO_4_ ZrO_2_, TiO_2_, Al_2_O_3_, ZrNb_2_O_7_	[65]
AlMo_0.5_NbTa_0.5_TiZr	1000	10	TiO_2_, Nb_2_O_5_, Ti_3_O_5_, ZrO_2_, Al_2_O_3_, MoO_3_	[71]
AlNbTiZr	600–1000	50	AlNbO_4_, Ti_2_ZrO_6_ Al_2_O_3_, NbO, ZrO_2_, TiO_2_	[61]
CrMoNbTaV	900–1100	25	Nb_2_O_5_, NbO_2_, CrTaO_4_, CrNbO_4_, Ta_9_VO_25_, Nb_9_VO_25_	[67]
AlNbTiVZr_0.25_	600–900	100	V_2_O_5_, VO_2_, TiO_2_, Nb_2_O_5_, TiNb_2_O_7_, AlNbO_4_, Nb_2_Zr_6_O_17_, ZrO_2_	[62]

## Data Availability

The data presented in this study are available on request from the corresponding author.

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
