# Peer review of "High-Temperature Oxidation of High-Entropic Alloys: A Review"

_materials, 2021, doi:10.3390/ma14102595_

Round 1
Reviewer 1 Report
High-temperature oxidation resistance is an important property for HEAs. This review manuscript can give some guides to develop heat-resistant alloy. It will be better if the authors can further clarify the following points.
- line 49-50: For the statement “The study of HEA oxidation is useful for analyzing the formation of HEAs, analyzing the high-temperature gas corrosion of these alloys, and analyzing the processes of obtaining oxide dispersion-strengthened alloys (ODS) in oxidizing HEAs. “, what is the relationship between oxidation and ODS?
- It seems that the effect of alloying elements on the oxidation resistance of HEAs is similar to that of conventional alloy. The authors should give more reviews or comments on the effect of special microstructure of HEAs on oxidation resistance.
- The authors gave mainly examples of high temperature atmosphere oxidation, it will be better also discuss oxidation in high temperature water contained (such as steam) environments, as this is a typical service environment.
Author Response
Dear Reviewer!
First of all, we would like to thank you for your attention to our work and your valuable recommendations for improving it. We have made the appropriate changes to the text of the manuscript. All fixes are highlighted in yellow.
We also want to additionally respond to your comments:
- line 49-50: For the statement “The study of HEA oxidation is useful for analyzing the formation of HEAs, analyzing the high-temperature gas corrosion of these alloys, and analyzing the processes of obtaining oxide dispersion-strengthened alloys (ODS) in oxidizing HEAs. “, whatistherelationshipbetweenoxidationand ODS?
Answer: You are absolutely right, in this manuscript there is no review of the data on obtaining dispersion-strengthened HEAs, therefore we have removed this phrase from the introduction
- It seems that the effect of alloying elements on the oxidation resistance of HEAs is similar to that of conventional alloy. The authors should give more reviews or comments on the effect of special microstructure of HEAs on oxidation resistance.
Answer: In conventional alloys, oxidation resistance can be greatly improved by the addition of a suitable (usually small) amount of Al, Cr or Si, as these elements can form a dense and stable oxide layer on the surface at high temperatures . HEAs based on metals of the Fe subgroup have fewer compositional restrictions than conventional structural alloys (e.g. stainless steels, Ni-based alloys), and may contain higher concentrations of elements such as Al, Cr and Si, which are necessary to form external protective oxide films. A HEAs with a high content of refractory metals is quite difficult to compare with conventional alloys. Their behavior at high temperatures is multidirectional and depends on the combination of metals used to obtain HEAs.
- The authors gave mainly examples of high temperature atmosphere oxidation, it will be better also discuss oxidation in high temperature water contained (such as steam) environments, as this is a typical service environment.
Answer: There are not many data in the literature about high-temperature oxidation of HEAs, and data on corrosion in a gaseous medium containing water vapor are even less. But our review contains such datas. For example, “Liu et al. [27] studied the oxidative behavior of high-entropy AlxCoCrFeNi alloys (x = 0.15; 0.4) in supercritical water ".
But, thanks for the recommendation, we have included this in the conclusions.

Reviewer 2 Report
Review of paper no. materials-1202761 titled High temperature oxidation of high-entropic alloys: a review by S. Veselkov et al.
This is an interesting review paper discussing an important topic. High entropy alloys (HEAs) constitute a new alloy concept with equimolar concentrations of 5+ constituent elements. The concept has been extensively developed over a past decade. As such, a review paper on high temperature oxidation of HEAs is needed by the community. The present paper is worthy of publishing subject to major revision.
1.The paper is extremely lengthy (34 pages). Furthermore, it does not provide a critical overview. It only reproduces original figures from previously published papers and describes them. Some critical approach to the subject matter is required.
2.The paper does not introduce the topic. The introduction is extremely short. Please, start with thermodynamics of metal oxidation. Introduce the thermodynamic stability of oxides (Elligham diagram) and discuss their Pilling-Bedworth ratios.
3.The paper basically studies 2 different alloy systems: CoCrFeNi-TM (TM=Al or Mn) and refractory HEAs. In most of these systems a layer of Cr2O3 and Al2O3 is formed. Therefore, the authors should introduce binary and ternary chromia and alumina-forming systems. Afterwards, they should discuss the effects of third, fourth and fifth elements. What is the critical concentration of Al and Cr in HEAs required to form a complete external scale? How does it compare with binary and ternary systems? Effects of rare earth elements and refractory metals on alumina and chromia scale formation should be discussed.
4.Many figures are abundant. They should be either removed or replaced with simple schematics of oxide scale composition. Please, remove abundant images showing the same thing several times (Fig. 4-8; 14-16). Only highlights should be reproduced from original papers. Other images should be replaced with simple schematics.
5.Parabolic rate constants should be extracted from weight gain data and compared. Lengthy tables 1, 3 and 5 should be converted into kinetic plots (see, e.g., Fig. 13 in the paper).
6.If weight gain data are available at several temperatures, Arrhenius plots should be provided. Activation energies should be compared.
7.Chapter 6 – new horizons in the development of HEAs – is very short. I feel the authors are unable to provide meaningful perspectives for the alloy development based on high temperature oxidation studies alone. This chapter should be removed. Relevant conclusions should be moved to the conclusions section.
8.Real conclusions start with point 4.1 (line 661). First three points should be removed.
End of comments
Author Response
Dear Reviewer!
First of all, we would like to thank you for your attention to our work and your valuable recommendations for improving it. We have made the appropriate changes to the text of the manuscript. All fixes are highlighted in yellow.
We also want to additionally respond to your comments:
1.The paper is extremely lengthy (34 pages). Furthermore, it does not provide a critical overview. It only reproduces original figures from previously published papers and describes them. Some critical approach to the subject matter is required.
Answer: the corresponding corrections were made to the text of the manuscript.
2.The paper does not introduce the topic. The introduction is extremely short. Please, start with thermodynamics of metal oxidation. Introduce the thermodynamic stability of oxides (Elligham diagram) and discuss their Pilling-Bedworth ratios.
Answer: the corresponding corrections were made to the text of the manuscript.
3.The paper basically studies 2 different alloy systems: CoCrFeNi-TM (TM=Al or Mn) and refractory HEAs. In most of these systems a layer of Cr2O3 and Al2O3 is formed. Therefore, the authors should introduce binary and ternary chromia and alumina-forming systems. Afterwards, they should discuss the effects of third, fourth and fifth elements. What is the critical concentration of Al and Cr in HEAs required to form a complete external scale? How does it compare with binary and ternary systems? Effects of rare earth elements and refractory metals on alumina and chromia scale formation should be discussed.
Answer: Direct data on oxide state diagrams will overload the manuscript and significantly increase its volume. On other issues, the corresponding changes have been made to the manuscript.
4.Many figures are abundant. They should be either removed or replaced with simple schematics of oxide scale composition. Please, remove abundant images showing the same thing several times (Fig. 4-8; 14-16). Only highlights should be reproduced from original papers. Other images should be replaced with simple schematics.
Answer: the corresponding corrections were made to the text of the manuscript.
5.Parabolic rate constants should be extracted from weight gain data and compared. Lengthy tables 1, 3 and 5 should be converted into kinetic plots (see, e.g., Fig. 13 in the paper).
Answer: We tried to rebuild the data from tables into graphs, but the resulting figures turned out to be overloaded with information, not readable and difficult for the researcher to perceive.
6.If weight gain data are available at several temperatures, Arrhenius plots should be provided. Activation energies should be compared.
Answer: the corresponding corrections were made to the text of the manuscript.
7.Chapter 6 – new horizons in the development of HEAs – is very short. I feel the authors are unable to provide meaningful perspectives for the alloy development based on high temperature oxidation studies alone. This chapter should be removed. Relevant conclusions should be moved to the conclusions section.
Answer: the corresponding corrections were made to the text of the manuscript.
8.Real conclusions start with point 4.1 (line 661). First three points should be removed.
Answer: the corresponding corrections were made to the text of the manuscript.

Reviewer 3 Report
The authors reviewed the high temperature oxidation behaviour of HEA alloys, which is no doubt a great interest to a wide audience especially researchers and engineers who are working on metallurgy. However, the manuscript was prepared in a low standard with a problematic structure and poor writing language. The reviewer finds it difficult to learn anything useful from this work and is therefore reluctant to recommend it for publication due to the following reasons.
- There is no doubt that a significant output of scientific publications on HEA alloys have become available during the last decade. Literature reviews should lead to new scientific insights and highlight gaps, conflicting results and under-examined areas of research. This paper simply listed the experimental results from other people's works (even including full tables on measurements of oxidation weight gains). This obviously will not save time for the scientific community who would rather read the original publications.
- The authors failed to provide a logic structure. No explanation was provided either on how the main body was divided in to 7 individual sections: introduction, Al-Co-Cr-Fe-Ni System, Mn-Co-Cr-Fe-Ni System, Refractory Metals, Oxidation products, New perspectives, conclusions. The reviewer does not see any correlation between each section, and the whole structure collapsed from the beginning.
- The manuscript also requires extensive editing of English language. There are countless grammar mistakes in the abstract that immediately took away the reviewer's enthusiasm in reading this work. For example, "interest in (not to)", "aimed to study (not studying)", etc. The careless paragraphing makes it even worse and the reviewer finds it very hard to follow the authors' writing.
- There are quite a few review papers on this topic, and even specifically for high temperature applications. https://doi.org/10.1016/j.jallcom.2018.05.067 So it is hard to agree with the authors that “there are practically no works...".
Author Response
Dear Reviewer!
First of all, we would like to thank you for your attention to our work and your valuable recommendations for improving it. We have made the appropriate changes to the text of the manuscript. All fixes are highlighted in yellow.
- There is no doubt that a significant output of scientific publications on HEA alloys have become available during the last decade. Literature reviews should lead to new scientific insights and highlight gaps, conflicting results and under-examined areas of research. This paper simply listed the experimental results from other people's works (even including full tables on measurements of oxidation weight gains). This obviously will not save time for the scientific community who would rather read the original publications.
Answer: the corresponding corrections were made to the text of the manuscript.
- The authors failed to provide a logic structure. No explanation was provided either on how the main body was divided in to 7 individual sections: introduction, Al-Co-Cr-Fe-Ni System, Mn-Co-Cr-Fe-Ni System, Refractory Metals, Oxidation products, New perspectives, conclusions. The reviewer does not see any correlation between each section, and the whole structure collapsed from the beginning.
Answer: the corresponding corrections were made to the text of the manuscript.
- The manuscript also requires extensive editing of English language. There are countless grammar mistakes in the abstract that immediately took away the reviewer's enthusiasm in reading this work. For example, "interest in (not to)", "aimed to study (not studying)", etc. The careless paragraphing makes it even worse and the reviewer finds it very hard to follow the authors' writing.
Answer: We have made work on the bugs
- There are quite a few review papers on this topic, and even specifically for high temperature applications. https://doi.org/10.1016/j.jallcom.2018.05.067 So it is hard to agree with the authors that “there are practically no works...".
Answer: this article is included in the text of the manuscript and in the list of references.

Round 2
Reviewer 2 Report
Authors mostly followed my comments. The manuscript has been improved. The paper is publishable subject to minor revision.
1.Original figures of alloys with variable Al concentration are abundant (Figs. 3-5, 19-21). They should be removed or replaced with simple schematics to save the journal space.
2.It is quite clear that a sufficient Al concentration (~20 at. %) is necessary to form a complete alumina (Al2O3) scale. The threshold for the complete alumina scale formation should be compared with simple binary systems, see, e.g., Materials (2020), 13, 3152, DOI: 10.3390/ma13143152.
3.Tables with weight gain data (Tables 1, 3 ,5) are still quite complex. It is advisable to analyze the raw data by parabolic rate law. The weight gains should be replaced with parabolic rate constants, if possible, to save the journal space.
4.Table 6 should be reduced as it runs over 3 pages. The alloys should be organized according to their oxidation products. Alloys with identical corrosion products should be grouped together. It is not necessary to specify the oxidation temperature and time for each alloy. The notes should be also removed.
End of comments
Author Response
Dear Reviewer, thank you for your valuable advices. We have made changes to the text of the manuscript and would like to respond to your comments:
1.Original figures of alloys with variable Al concentration are abundant (Figs. 3-5, 19-21). They should be removed or replaced with simple schematics to save the journal space.
Answer: Corresponding corrections were made to the text of the manuscript. Instead of fig. 3-5 now there is only fig. 3, instead of Fig. 19-21 now there is only fig. 17. We also removed two tables, now there are only 5 of them, which made it possible to reduce the number of pages in the manuscript.
2.It is quite clear that a sufficient Al concentration (~20 at. %) is necessary to form a complete alumina (Al2O3) scale. The threshold for the complete alumina scale formation should be compared with simple binary systems, see, e.g., Materials (2020), 13, 3152, DOI: 10.3390/ma13143152.
Answer: Corresponding corrections were made to the text of the manuscript. The list of references is supplemented:
«Šulhánek P. et al. [73] were found that for the formation of a protective film consisting only of Al2O3, the required concentration of aluminum in the alloy must be more than 20 at. %. For the presented in Table 5 alloys, the Al concentration does not reach this indicator; therefore, the composition of the observed oxide films is very complex and varied.
- Šulhánek, P.; Drienovský, M.; Ĉerniĉková, I.; Ďuriška, L.; Skaudžius, R.; Gerhátová, Ž.; Palcut, M. Oxidation of Al-Co alloys at high temperatures. Materials. 2020, 13, 3152. doi: 10.3390/ma13143152»
3.Tables with weight gain data (Tables 1, 3 ,5) are still quite complex. It is advisable to analyze the raw data by parabolic rate law. The weight gains should be replaced with parabolic rate constants, if possible, to save the journal space.
Answer: You propose to include information from the table in one picture. But, unfortunately, putting all the data on one figure will make it unreadable and difficult to understand. Splitting information into several pictures will also not make the information easier to perceive and will additionally increase the number of pages in the manuscript. Sorry, but we think that presentation this datas in form of table is more convenient for readers.
4.Table 6 should be reduced as it runs over 3 pages. The alloys should be organized according to their oxidation products. Alloys with identical corrosion products should be grouped together. It is not necessary to specify the oxidation temperature and time for each alloy. The notes should be also removed.
Ответ: Мы переделали стол. Проведенный нами анализ литературы позволил составить перечень продуктов окисления ВОА, обнаруженных в экспериментах (табл. 5). Данные представлены по мере усложнения состава образующейся оксидной пленки.
Конец комментариев
Все исправления во втором раунде выделены синим цветом.

Reviewer 3 Report
This response fails to refer explicitly to the previous and revised versions of the manuscript and explain what changes have been made. The reviewer does not see a significant improvement in the revised manuscript either.
Author Response
Dear Reviewer, we apologize that in our first reply we were unable to reflect all the changes we made in accordance with your recommendations. Since in the second round your comments remained the same, we will try to answer them in more detail:
(All corrections made in accordance with your and other reviewer’s comments are highlighted in yellow (round 1) and blue (round 2) bullets).
- There is no doubt that a significant output of scientific publications on HEA alloys have become available during the last decade. Literature reviews should lead to new scientific insights and highlight gaps, conflicting results and under-examined areas of research. This paper simply listed the experimental results from other people's works (even including full tables on measurements of oxidation weight gains). This obviously will not save time for the scientific community who would rather read the original publications.
Answer: the corresponding corrections were made to the text of the manuscript. This remark largely coincides with the wishes of other reviewers.
We have significantly revised the text of the manuscript. The number of figures was reduced to 17, and the number of tables to 5. Since you are absolutely right and the superfluous descriptive part had to be removed.
In summary tables 1, 3, 4, the activation energies calculated by us on the basis of the literature data were added, and the data of tables 1, 3, 4, 5 were grouped in a certain way.
A discussion of literature data was added to the text of the manuscript.
«Most of the articles presented in the literature describe the high-temperature oxidation of HEAs of this group in air, with the exception of the work of Liu et al. [27], which describes the corrosion resistance of alloys at high pressures in an atmosphere containing water vapor. Since there is practically no data in the literature on the behavior of HEAs under such conditions, we are obliged to dwell on the description of work [27] in more detail.
Among other data presented in the literature on high-temperature gas corrosion in air for the HEAs based on the Alx-Co-Cr-Fe-Ni system, the most complete study, in our opinion, was carried out by a group of researchers, Butler et al. [28–30].»
«After analyzing the data from Table 1, it can be noted that the best results for the base alloy were obtained for the composition Al30(CoCrFeNi)70 [28]; copper additives increase corrosion resistance (AlCoCrCuFeNi composition [33]); and the alloy with the addition of silicon and without iron in its composition showed the greatest resistance (Al3Co2Cr7Ni7Si [30]). At the same time, all three alloys are characterized by a protective oxide film based on Al2O3 with the possibility of forming a layer of Cr2O3.
It should also be taking in account that in the work of Lu et al. [41] there is an extremely low corrosion rate of the alloy AlCoCrFeNiYHf (kp = 1.9×10–13 g2 cm–4 s–1), which probably explains the lack of data on weight gain. Thus, the additions of yttrium and hafnium have a positive effect on the corrosion resistance of HEAs based on the Al-Co-Cr-Fe-Ni system.
It can be noted that literature data are mainly of an empirical nature and only represent the results of experiments in the form of a technical report. As such, there is no analysis of the mechanism of the effect of adding various elements to the base alloy Al-Co-Cr-Fe-Ni on the process of high-temperature gas corrosion. In this connection, the question arises about the gaps in the presented approaches to the study, in particular, it is necessary to use the tools of thermodynamic modeling to describe the dependence of the phase composition of the formed corrosion products from the alloy composition and oxygen pressure in the system. There is also insufficient data on the kinetic laws of high-temperature oxidation of HEAs based on the Al-Co-Cr-Fe-Ni system.
Thus, there is no clearly defined line of research in the literature. The influence of hafnium additives seems promising, which leads to the idea of a possible positive effect of tantalum or tungsten additives. It also makes sense to consider the complex effect of doping with copper in conjunction with gold, silver and platinum (according to the data from Ellingham diagram).»
«The most thoroughly the resistance to high-temperature gas corrosion of HEAs this group of alloys was studied in works [42, 43]. Let's dwell on the results of these works in more detail.»
«Analyzing the data from the Table 3, we note that the alloy showed the best corrosion resistance is Mn0.5Co26Cr22Fe25Ni26 [42], oxide film on which consisted only of Cr2O3. This is consistent with the calculation of the activation energy, according to which the rate of the oxidation reaction for this alloy will be the lowest of all. Yttrium additions did not affect on the corrosion resistance of the presented alloys».
«Mn-Co-Cr-Fe-Ni alloys seem less attractive in terms of their corrosion resistance compared to the alloys in the previous chapter. Comparing Table 1 and Table 3, it can be noted that even the test temperature for Mn-Co-Cr-Fe-Ni alloys is lower by 300–400 degrees than for testing the Al-Co-Cr-Fe-Ni HEAs. On the other hand, Mn-Co-Cr-Fe-Ni HEAs have not been sufficiently studied and the effect of additional elements has not been considered in practice, in particular, it can be assumed that the addition of aluminum would contribute to the formation of a continuous protective oxide film on the surface of these alloys.»
«There are a number of studies of the behavior of refractory HEAs during high-temperature oxidation [49, 53–71]. The most interesting results are given below, especially the role of aluminum in oxidation resistance should be noted».
«According to the data from the Table 4, the best corrosion resistant is available for the alloys HfNbTiZr [61], AlCrMoTaTi [66] and AlTaMoCr [70]. Moreover, as noted by researchers [66, 70], it is the combination of molybdenum and tantalum that makes it possible to obtain such excellent properties. All three alloys have a varied composition of the oxide film. For the HfNbTiZr alloy, the oxidation products contain NbO, ZrO2, TiO2; for the AlCrMoTaTi alloy – Al2O3, Cr2O3, TiO2, CrTaO4, for the AlTaMoCr alloy - Cr2O3, Al2O3, CrTaO4. Note that, for the last two compositions, Mo compounds are absent in the oxidation products.
A variety of compositions of refractory HEAs can be noted, which speaks more of a certain haphazard development of compositions than a logically built chain of experiments. It can also be immediately noted that HEAs, consisting entirely of only refractory metals, have a relatively low resistance to high-temperature gas corrosion. The situation changes as soon as aluminum and / or chromium are additionally introduced into the composition.
Thus, it seems that the creation of alloys based on the basic composition Al-Co-Cr-Fe-Ni with the addition of refractory elements (for example, the combination of tantalum with molybdenum, hafnium or zirconium) is perspective trend for investigators.
The summary information given in Table 4 is supplemented with the activation energy calculated by us. The obtained values of the activation energy correlate well with the experimental data on the change in the mass of the samples - materials with a high activation energy have a high resistance to oxidation».
The text of the conclusion was supplemented with our comments of the worked out literary material, as well as missing information on high-temperature gas corrosion.
«This review and analysis of data on the oxidative behavior of HEAs leads to the following conclusions.
(1) The experimental data on the phase formation and phase composition of the oxide films during the oxidation of a number of widely studied HEAs have been collected and analyzed. These data can be used to test theoretical models of the high-temperature oxidation of HEAs.
(2) Promising heat- and corrosion-resistant HEAs can be found in alloys of the Al-Co-Cr-Fe-Ni system. The addition of Si, Ti, Y, and Hf makes it possible to increase the resistance to high-temperature gas corrosion of such alloys. The effect of Zr and Ta additives requires additional research. Mo additives have a negative effect on corrosion resistance, since Mo oxides are characterized by low evaporation temperatures. And even at low concentrations of Mo, the presence of Mo oxides in the oxide layer can lead to the discontinuity of the scale, and make it loose and porous. The effect of Cu additions also requires additional research, but it is already clear that at sufficiently high Cu concentrations (when its amount is comparable with the amounts of other elements forming a multicomponent base) the adhesion of the resulting oxide layer with respect to the matrix surface is low; such scale peels off and breaks easily.
(3) HEAs of the Mn-Co-Cr-Fe-Ni system are less resistant to high-temperature gas corrosion than alloys without Mn. The mechanism of oxidation of such alloys still needs to be studied, however, it is known that a multiphase, discontinuous scale is formed on such alloys, while the scale layers containing Mn oxide are porous, peel off easily, and do not interfere with the further oxidation of the alloy.
(4) Studies of the resistance to high-temperature corrosion of HEAs based on refractory metals are contradictory. According to the literature, the alloys most resistant to high-temperature oxidation are the HEAs of the Al-Cr-Mo-Ta-Ti system. In the presence of Ta, Mo does not participate in the oxidation process and, therefore, there is no Mo oxide in the scale. In the absence of Al, oxides of refractory metals do not have sufficient protective properties.
(5) The results of the experimental work presented in the literature allowed us to come to the following empirical conclusions about possible composition of corrosion-resistant at high temperature HEAs:
(5.1) The most important elements that contribute to an increase in the resistance of HEAs to oxidation are Al and Cr. Continuous films of Al2O3 or Cr2O3 with high adhesion to the alloy can be formed, which prevent the directional diffusion of oxygen into the alloy matrix. These same elements can react with other elements in the HEA (for example, Fe and Ni), which can lead to the formation of spinel films, which also increase resistance to oxidation. Elements such as Si and Y can also improve oxidation resistance.
(5.2) The presence of refractory elements, such as W, Mo, Ta, Nb, V, Ti, Zr and Hf, can increase the strength of HEAs at high temperatures, but some of these elements (primarily Mo, W, and V) can accelerate the oxidation of alloys, primarily due to the formation of volatile oxides.
(5.3) When Ti is one of the main elements of an alloy it plays an important role in the formation of protective layers from CrTaO4, TiO2, Ti3O5 and ZrO2 oxides, while simultaneously reducing the number of oxides that reduce heat resistance (Nb2O5, Ta2O5).
(6) Thus, it seems that the creation of alloys based on the basic composition Al-Co-Cr-Fe-Ni with the addition of refractory elements (for example, the combination of tantalum with molybdenum, hafnium or zirconium) is perspective trend for investigators.
(7) Several gaps can be noted in the study of high-temperature oxidation of HEAs:
(7.1) Thermodynamic modeling tools (for example, CALPHAD) are practically not used to predict high-temperature oxidation processes. The use of thermodynamic modeling techniques would optimize the search for compositions of corrosion-resistant HEAs.
(7.2) All studies of high-temperature oxidation of HEAs are empirical. In the literature, there are no theoretical data on the mechanism of the formation of protective films on HEAs and the effect of additional elements on this mechanism.
(7.3) For the study of high-temperature gas corrosion of HEAs, the authors proposed practically only a single method of oxidation in air. But for practical use it would be useful to conduct tests in the environment H2/H2O (water vapor); in the environment SO2/SO3 (typical for the oil and gas industry) or in the environment CO/CO2 when engines are running in aircraft and rocketry mechanisms.»
- The authors failed to provide a logic structure. No explanation was provided either on how the main body was divided in to 7 individual sections: introduction, Al-Co-Cr-Fe-Ni System, Mn-Co-Cr-Fe-Ni System, Refractory Metals, Oxidation products, New perspectives, conclusions. The reviewer does not see any correlation between each section, and the whole structure collapsed from the beginning.
Answer: We have significantly revised the introduction. We substantiated the choice of systems for research:
«The Ellingham diagram [14, 15], which shows the change in the Gibbs energy of the formation of metal oxides depending on the temperature and composition of the gas phase, can give some idea of the possible oxidative behavior of HEAs at high temperatures.
HEAs based on light elements and lanthanides, taking into account the properties of the elements that form them, are oxidized easily and completely. On the other hand, the possibility of high-temperature oxidation of platinum-group metals at pressures close to atmospheric is hardly worth studying.
If we analyze the data of the Ellingham diagram [14, 15], as well as the data of the Pilling-Bedworth ratio [16] (which is a criterion for the continuity of the formed oxide film), we can conclude that interesting data of oxidation can be observed in HEAs based on metals of the Fe subgroup and HEAs based on refractory transition metals. Therefore, in this review, we analyze works devoted to the high-temperature oxidation of these groups of HEAs.
It should be noted that among the systems based on the Fe subgroup, the most studied are the Al-Co-Cr-Fe-Ni and Mn-Co-Cr-Fe-Ni systems. It is possible that this is due to the fact that the Al and Cr in their composition, which according to the Ellingham diagram are oxidized better than other components of these alloys (i.e. the formation of oxides of these elements is thermodynamically more likely) and have a Pilling-Bedworth ratio more than one, imparting protective properties to films made of oxides based on them».
Next, we have described the structure of the review:
«This review consists of six chapters. The first chapter is introduction to the problem. The second and third chapters are devoted to a review of the oxidative behavior of HEAs based on the Fe subgroup, namely, the most common and studied systems of HEAs are Al-Co-Cr-Fe-Ni and Mn-Co-Cr-Fe-Ni, respectively. The systems were divided into separate chapters, not only for the convenience of readers, but also because of a fairly large amount of literature data on each of the systems. The fourth chapter is devoted to a review of the behavior during high-temperature oxidation of HEAs based on refractory metals. The fifth chapter summarizes the data on the oxidation products of all considered alloys, and also considers the possibility of predicting the phase composition of the products of high-temperature gas corrosion using thermodynamic modeling techniques. The need to include data in the fifth chapter is due to the accepted convenience of reading the review article and the use of data from different sources. The sixth chapter is a logically built chain of conclusions, taken on the basis of all the considered literature data on high-temperature oxidation of HEAs».
At the request of one of the reviewers, we removed one of the chapters (perspectives) and, in this connection, reworked the "Conclusion" section.
- The manuscript also requires extensive editing of English language. There are countless grammar mistakes in the abstract that immediately took away the reviewer's enthusiasm in reading this work. For example, "interest in (not to)", "aimed to study (not studying)", etc. The careless paragraphing makes it even worse and the reviewer finds it very hard to follow the authors' writing.
Answer: We have made work on the bugs.
- There are quite a few review papers on this topic, and even specifically for high temperature applications. https://doi.org/10.1016/j.jallcom.2018.05.067 So it is hard to agree with the authors that “there are practically no works...".
Answer: this article is included in the text of the manuscript and in the list of references.
«Существуют исследования, обобщающие самые разнообразные аспекты образования ВЭУ [7-12], но практически нет обзоров, посвященных окислению ВЭУ при повышенных температурах, хотя в этой области получено достаточно большое количество информации, за исключением работы Chen et al. [13], где рассматриваются только некоторые аспекты высокотемпературного окисления тугоплавких ВЭУ (более того, за последние три года опубликовано достаточное количество новых данных).
- Chen, J.; Zhou, X.; Wang, W.; Liu, B.; Lv, Y.; Yang, W.; Xu, D.; Liu, Y. Обзор фундаментальных исследований высокоэнтропийных сплавов с перспективными высокотемпературными свойствами. Дж. Сплавы Компд. 2018, 760, 15-30. doi: 10.1016/j.jallcom.2018.05.067»

Round 3
Reviewer 3 Report
The reviewer would like to thank the authors for the revised manuscript. Significant changes were made to address the reviewer's comments.
Just a few more comments:
- It remains unclear why Table 4, i.e. weight gain, should be kept even after the addition of the activation energy (Anyone can calculate this themselves). If the authors really want to keep this, these data should be plotted in a graph.
- Extensive editing of English language and style are still required, e.g., 1st sentence in abstract, "Interested in" is used when what comes after it is a noun, or a verb acting like a noun (known as a gerund). "Interested to" is used when what comes after it is a verb in its "to form" (known as an infinitive). [to do something].
- It is a bit irritating to read anything with so many short paragraphs. Most of them should be combined into one. The same problem exists in the Conclusion part, in which, the authors listed 10 lengthy points. The conclusion is intended to help the reader understand why your research should matter to them after they have finished reading the paper. A conclusion is not merely a summary of your points or a re-statement of your research problem but a synthesis of key points.
Author Response
Dear reviewer, we want to thank you again for your valuable comments and we want to respond to your remarks:
Just a few more comments:
- It remains unclear why Table 4, i.e. weight gain, should be kept even after the addition of the activation energy (Anyone can calculate this themselves). If the authors really want to keep this, these data should be plotted in a graph.
Answer: Activation energy was added by us in accordance with the comment of one of the reviewers. Converting tables to graphs was also a comment from one of the reviewers. We tried to do this, but the drawing turned out to be too overloaded with information and unreadable. Therefore, and the reviewer agreed with us, we left the data in the form of a table.
- Extensive editing of English language and style are still required, e.g., 1st sentence in abstract, "Interested in" is used when what comes after it is a noun, or a verb acting like a noun (known as a gerund). "Interested to" is used when what comes after it is a verb in its "to form" (known as an infinitive). [to do something].
Answer: When preparing the manuscript, we used the services of a professional translator. We have done proofreading again.
- It is a bit irritating to read anything with so many short paragraphs. Most of them should be combined into one. The same problem exists in the Conclusion part, in which, the authors listed 10 lengthy points. The conclusion is intended to help the reader understand why your research should matter to them after they have finished reading the paper. A conclusion is not merely a summary of your points or a re-statement of your research problem but a synthesis of key points.
Answer: Thanks for the advice. We have reduced the number of paragraphs in the text of the article. Also we shortened and reworked the conclusion.
Changes are highlighted in green.